# Performance of a low-cost optical particle counter (Alphasense OPC-N3) for estimating size-resolved dust emission flux using eddy covariance

Sylvain Dupont[1], Eric Lamaud[1], Mark R. Irvine[1], Jean-Marc Bonnefond[1], Adolfo Gonzalez-Romero[2,3], Andrés Alastuey[3], Cristina González-Flórez[2], Xavier Querol[3], Konrad Kandler[4], Martina Klose[5], and Carlos Pérez García-Pando[2,6]

[1]INRAE, Bordeaux Sciences Agro, ISPA, F-33140 Villenave d'Ornon, France
[2]Barcelona Supercomputing Center (BSC), 08034, Barcelona, Spain
[3]Institute of Environmental Assessment and Water Research (IDAEA-CSIC), 08034, Barcelona, Spain
[4]Institute of Applied Geosciences, Technical University of Darmstadt, 64287 Darmstadt, Germany
[5]Institute of Meteorology and Climate Research Troposphere Research (IMKTRO), Karlsruhe Institute of Technology (KIT), 76131 Karlsruhe, Germany
[6]ICREA, Catalan Institution for Research and Advanced Studies, 08010, Barcelona, Spain

**Correspondence:** Sylvain Dupont (sylvain.dupont@inrae.fr)

**Abstract.** The recent development of low-cost optical particle counters (OPCs) presents new opportunities for improving spatial coverage of particle concentration in the atmosphere as they are more affordable, compact, and consume less power than traditional OPCs. In particular, these OPCs could improve our ability to quantify dust emissions in complex environments during aeolian soil erosion. The high frequency sampling capacity (1 Hz) of some sensors may make them suitable for estimating dust emissions using the eddy-covariance method. Here, the capability of the low-cost OPC-N3 from Alfasense to estimate the size-resolved dust flux using the eddy-covariance method is evaluated. During the Jordan Wind Erosion and Dust Investigation (J-WADI) experiment, we tested one OPC-N3 against traditional reference OPCs. The OPCs were located in close proximity to a sonic anemometer, enabling the correlation of dust concentration and vertical velocity fluctuations for estimating dust fluxes. Despite the high temperature and dusty wind conditions of the campaign, the N3 monitored the dynamics and magnitude of dust concentration with reasonable precision. The turbulence characteristics of the dust concentration fluctuations measured by the N3 were similar to those from the reference OPC. After calibrating the N3 dust concentration, the N3 accurately estimated the dust emission flux, with differences of less than 30% compared to the reference OPC. Our results confirm the potential of low-cost OPCs for dust-erosion research. Nonetheless, further evaluation of low-cost OPCs is still needed across different environments and weather conditions.

## 1 Introduction

Mineral dust represents an important component of the particles present in the atmosphere. Its presence affects air chemistry, cloud formation, sunlight radiation reaching the earth's surface, and nutrient availability for ecosystems (Shao, 2008). As a consequence, mineral dust plays a crucial role in Earth's climate system (Petit et al., 1990; Tegen and Lacis, 1996; Mahowald

et al., 2014), as well as for the growth of land and oceanic organisms (Mahowald et al., 2011; Knippertz and Stuut, 2014). These environmental impacts are influenced by the amount and size range of dust particles in the air (Mahowald et al., 2014; Kok et al., 2017; Adebiyi and Kok, 2020), which, in turn, are closely related to the mechanism of dust release from and deposed on the ground.

Our comprehension of dust emission and deposition is partial because of the complexity of the involved processes, their intermittent nature, as well as the variety of erosive surfaces, and their spatial heterogeneity, especially in semiarid regions. Deploying enough optical particle counters (OPC) in complex environments to continuously measure size-resolved dust concentration is challenging. This limitation makes it difficult to determine dust fluxes with good spatial resolution. Estimating dust fluxes is crucial for quantifying dust sources and deposition, and for developing and evaluating dust emission-deposition parameterizations for global climate models. Focusing only on dust concentration is insufficient as concentration integrates both locally emitted particles and particles emitted far upwind from the surface.

Field experiments that estimated dust emission fluxes were mainly limited to two levels of expensive OPCs to evaluate dust fluxes at only one point from the vertical gradients of dust concentration (e.g., Gillette et al., 1972; Nickling and Gillies, 1993; Sow et al., 2009; Shao et al., 2011; Ishizuka et al., 2014; González-Flórez et al., 2023). This approach, known as the flux-gradient (FG) method, requires accurate measurements of dust concentration to produce a meaningful gradient. In the continuity of Fratini et al. (2007), Dupont et al. (2021) developed an eddy-covariance (EC) approach for estimating the near-surface dust flux by utilizing only one OPC sampling at 1 Hz. This approach derives the flux from the correlation at one point between the dust concentration fluctuations and the vertical velocity fluctuations, rather than a difference of two dust concentration levels. This more localized flux estimation makes it better suited to complex environments, allowing a fine investigation of surface heterogeneity impacts on dust fluxes at various scales (e.g., Dupont et al., 2024). This EC approach has been evaluated on soil erosion in Tunisia (Dupont et al., 2019; Dupont, 2020; Dupont et al., 2021; Dupont and Patton, 2022) and more recently used in Iceland and Jordan (Dupont et al., 2024). The EC approach opens new perspectives to more easily obtain a detailed spatial resolution of dust fluxes from a limited number of OPCs in order to better estimate the environmental impacts of dust events.

In recent years, low-cost OPCs have been developed to measure airborne particles ranging from submicron to several tens of microns in diameter. Here, low-cost refers to sensors costing around 50 to 1000€ compared to 20-25 k€ for traditional OPCs. Their affordability, small size and low consumption make them very attractive for monitoring airborne particle concentrations over complex environments with high spatial resolution. These OPCs are based on the same principles of light scattering property of particles as more expensive and bulkier traditional OPCs (Dubey et al., 2022). They detect particles individually, offering details about their quantity and size.

One of these low-cost OPC, the Alphasense OPC-N3 ($\approx$ 600€), possesses a frequency sampling capacity of 1 Hz, equivalent to the traditional OPC used by Dupont et al. (2021). This means that dust fluxes could be potentially estimated via the EC method from this OPC combined with a high-frequency anemometer. We used the opportunity of the Jordan Wind erosion And Dust Investigation (J-WADI) campaign [see https://www.imk-tro.kit.edu/english/11800.php and Dupont et al. (2024)] to install an OPC-N3, referred hereafter as N3, at the same level as two traditional OPCs, a Promo 2000 and a Fidas 200S, both

from Palas GmbH. The N3 and Promo OPCs were collocated with a high-frequency sonic anemometer, allowing to compare the estimated dust fluxes using the EC method.

Some comparative studies between Alphasense OPC-N series and reference OPCs have already shown promising performances, showing that low-cost OPCs can be suitable to monitor aerosols after a calibration correction (e.g., Crilley et al., 2018; Dubey et al., 2022; Nurowska et al., 2022). These comparisons were mainly done on particulate matter ($PM_{10}$ or $PM_{2.5}$) for air quality monitoring through indoor and outdoor experiments, especially in urban areas or nearby roads, for fog monitoring, less in the context of aeolian soil erosion and mineral dust emission (Kaur and Kelly, 2023). The goal of this study is to evaluate the ability of the low-cost Alphasense OPC-N3 to estimate the size-resolved dust flux using the EC approach, during the erosion events of the J-WADI campaign.

## 2 Methodology

### 2.1 Field experimental

The J-WADI campaign took place in the north of the Wadi Rum desert in Jordan, from September 9th to October 5th 2022. This campaign was co-organized by the FRontiers in dust minerAloGical coMposition and its Effects upoN climaTe (FRAGMENT) and Helmholtz Young Investigator Group "Mineral Dust" projects. The site was located between two gentle hills in the North and South, with a wind channeling most of the time between them (Figure 1a). The site between these two hills was flat without significant roughness elements, with an extended fetch in the main wind direction.

During the campaign, several measurements were performed to characterize aerosols and the meteorology. In particular, a 10 m-high mast (Figure 1) was equipped with 3D and 2D sonic anemometers and temperature sensors to measure the mean horizontal wind velocity and temperature profiles, momentum and heat fluxes, and flow turbulence main characteristics. For our purposes here, N3 and Promo 2000 OPCs were installed close to each other at 2 m height, with their sampling heads at approximately 20-30 cm distance behind a 3D sonic anemometer (Campbell® Scientific CSAT3), to compare the estimated EC dust fluxes from both OPCs (Figure 1b). At about 20 m from the mast, a Fidas 200S OPC was placed at 2 m high on a scaffolding. An additional Fidas 200S OPC was placed on the same scaffolding at 4 m high, but is not used here.

During a previous campaign in Iceland in 2021 (Dupont et al., 2024), seventeen N3 OPCs were intercompared during three days following their deployment over several months across the Vatnajökull National Park. This campaign was organized by the FRAGMENT and HiLDA (Iceland as a model for high-latitude dust sources – a combined experimental and modeling approach for characterization of dust emission and transport processes) projects. All N3 OPCs were placed at the same height and in close proximity (within a few meters). Despite the absence of notable erosion events during this intercomparison period, a short nighttime period with high particle concentrations, probably corresponding to fog droplets rather than mineral dust, permitted the investigation of the variability of particle concentrations between the N3 OPCs (section 3.5).

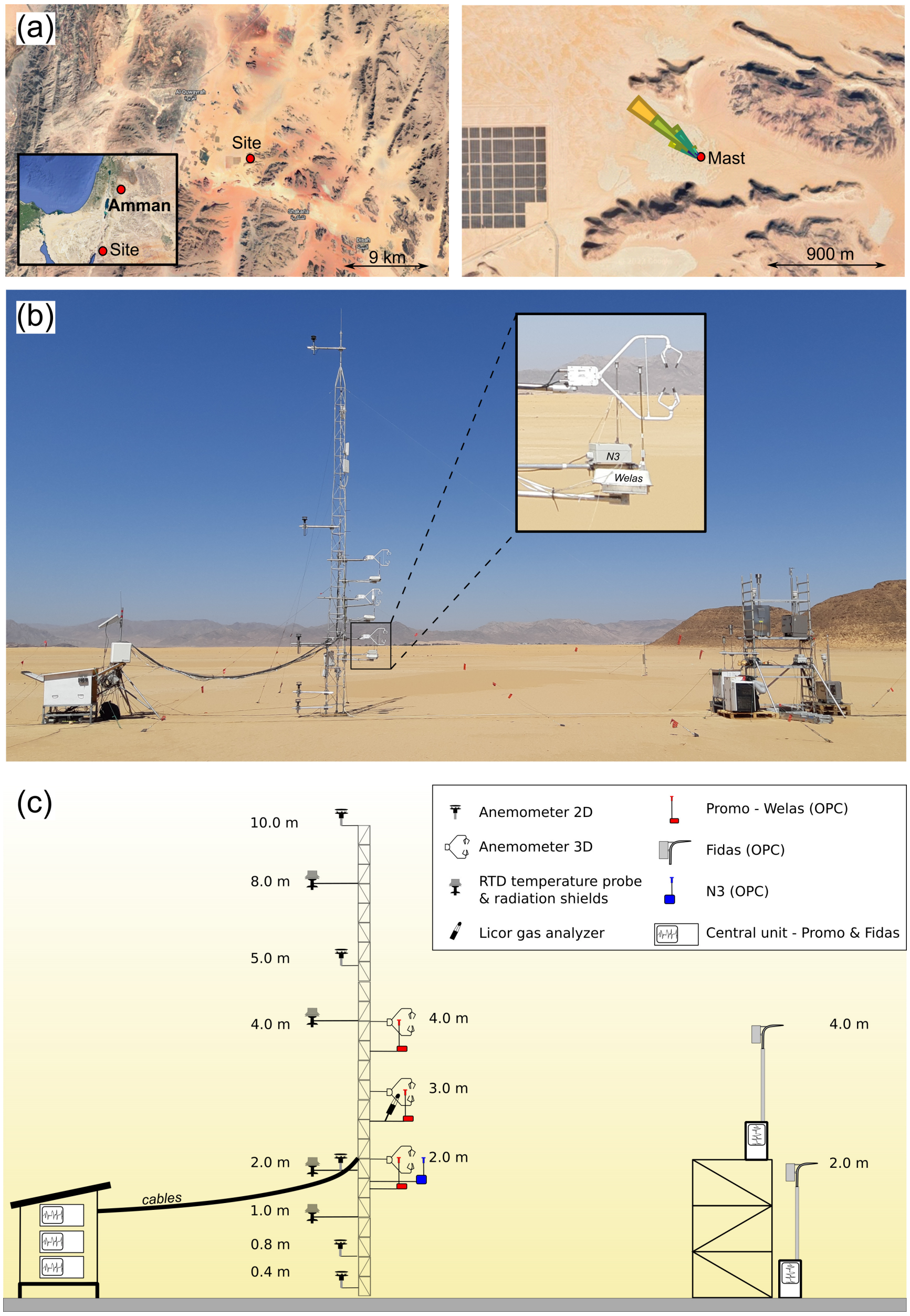

**Figure 1.** (a) J-WADI experimental site in Jordan, with the location of the 10 m high mast (red dot) and wind rose during the main erosion event (September 29, 2022). (b) Picture of the mast on which the sonic anemometers and N3 and Promo OPCs were mounted, among others, with a zoom on the 2-m high N3 and Welas (Promo) devices coupled to a sonic anemometer. The two Fidas OPCs were mounted on the scaffolding on the right at two levels. (c) Schematic description of the mast and scaffolding, with the sensors mounted on them. Satellite views are from © Google Maps.

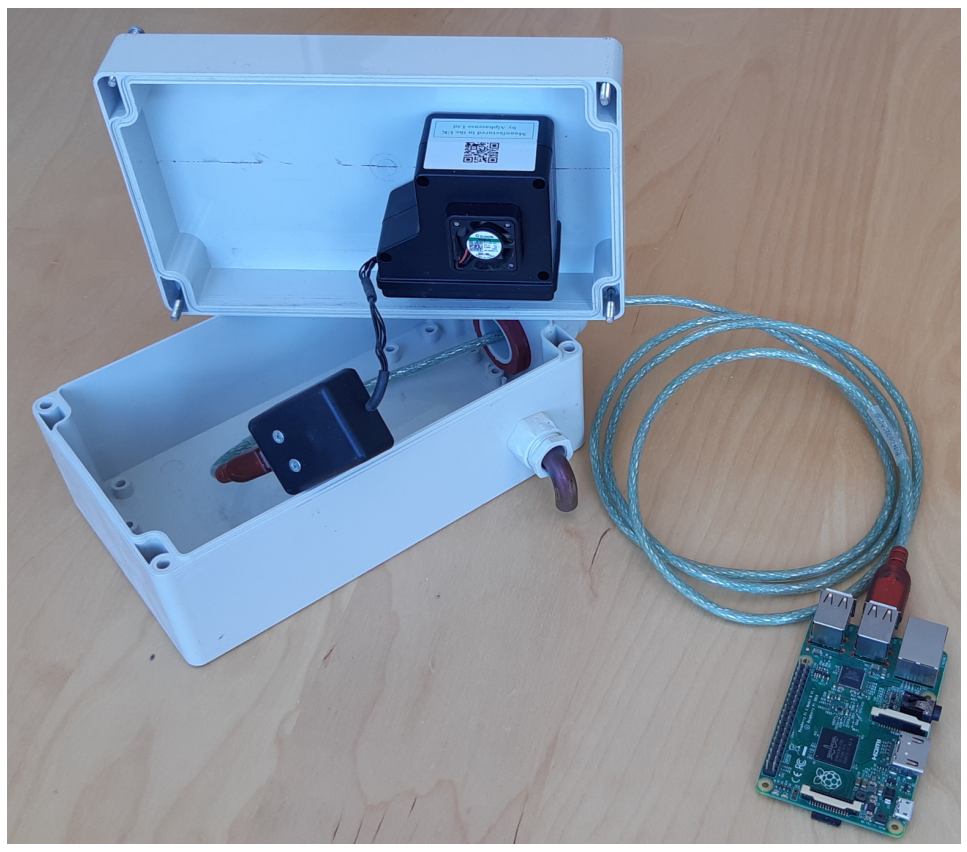

**Figure 2.** Picture of the OPC-N3 inside the waterproof box visible in Figure 1b, connected to a Raspberry Pi for data acquisition. An air outlet is located on one side of the box.

### 2.2 Optical particle counters

85 The low-cost N3 OPC from Alphasense was evaluated for its ability to measure dust concentration magnitude and fluctuations to derive afterward an EC dust flux. The N3 is a miniaturised OPC (75×63.5×60 mm, Figure 2) that measures particle concentration from 0.35 to 40 $\mu$m divided in 24 bins, with a sampling rate up to 1 Hz. For outdoor use, the N3 was installed inside a small home-made waterproof plastic box (see Figures 1b and 2). Conversely to traditional OPCs, it does not have a pump but a small fan aspirator, which explains its low consumption (0.9 W). The sensor measures up to 10,000 particles $s^{-1}$. At each time step, the number of detected particles $n_b$ per size bin $b$ were converted into particle concentration $C_b^{N3}$ (in particles $m^{-3}$) as follows: $C_b^{N3} = n_b/(F_r \times T_s)$, where $F_r$ is the flow rate and $T_s$ is the sampling time. During the J-WADI campaign, $T_s$ was nearly constant with a value around 0.46 s for an acquisition period of 1 s, meaning that half of the acquisition time step was used for measurements. The flow rate $F_r$ was about 0.16 $\mathrm{L\,min}^{-1}$ but with some fluctuations (see Section 3.2).

The Promo 2000 OPC from Palas GmbH was used as a reference OPC to evaluate the N3 OPC. This OPC is a light-scattering aerosol spectrometer system for particle size analysis and concentration determination that was equipped with the aerosol sensor Welas 2300 sampling up to 1 Hz as the N3. This Welas sensor was installed on the mast and covered by a white metallic box for outdoor protection. This sensor was connected to the Promo 2000 control unit through two 10 m long fiber optic cables that transmitted white light to the measuring volume and received the measured scattered-light pulse. In addition, a 10 m long flexible tube connected the Welas sensor to the pump located inside the control unit. The Promo 2000 central unit was located in a small wooden structure nearby the mast. The welas sensor measures number concentration up to 40,000 particles $\mathrm{cm}^{-3}$. Its consumption is about 100 W. For this experiment, the Welas measured particles in the range of 0.3-17 $\mu$m with 32 intervals per decade. The spectrometers determined the size and number of particles in sampled air in the optical chamber, delivered by a pump with a stable flow rate of 5 $\mathrm{L\,min}^{-1}$ (see Section 3.2).

Both the N3 and the Promo were equipped with the same small sampling heads especially designed (a) to minimize disruption of the air near the sonic anemometers while sampling dust particles within the air, and (b) to minimize the time-lag between wind and dust measurements (Dupont et al., 2021). This head consisted of a 45 cm long and 0.5 cm diameter tube with a drilled cover that allows particles to enter while protecting the inlet from rain. Although this sampling head is well suited to perform eddy covariance, it underestimates the concentration of coarse particles ($> 5\,\mu$m in diameter) (Dupont et al., 2021).

The Fidas 200S from Palas GmbH is a certified optical analyser for continuous outdoor monitoring of PM2.5 and PM10 with a sampling frequency up to 1 Hz. The Fidas is mounted in a splash-proof stainless steel control cabinet for outdoor use. The sampling system of the Fidas operates with a volume flow of 4.8 $\mathrm{Lmin^{-1}}$. Its normal consumption is about 60 W. The measuring range was 0.4 to 40 $\mu$m with 32 intervals per decade during the J-WADI campaign. The sensor measures number concentration up to 20,000 particles $\mathrm{cm^{-3}}$. Compared to the N3 and Promo, the Fidas was equipped with a conical sampling head with a 4.5 mm entry diameter, designed for near-isokinetic operation at a wind speed of 5 $\mathrm{ms^{-1}}$. To improve collection efficiency for larger particles, the sampling head is fitted with a wind vane, allowing it to continuously align with the wind direction.

Data from the three OPCs and sonic anemometers were acquired and stored simultaneously from synchronized Raspberry Pi 3 assembled in a local network from a python script. Each OPC had its own Raspberry Pi.

For this study, the three OPCs assume spherical particles with a density of 1650 $\mu\mathrm{gm^{-3}}$, and a refractive index of $1.50 + 0i$ for the N3 (default value) and $1.59 + 0i$ for the Promo and Fidas (latex value).

## 3 Results and discussion

### 3.1 Long term dust concentration measurements

Figure 3 presents the time series of the 15-min averaged size-resolved airborne particle concentration in number measured by the three OPCs during 23 consecutive days of the J-WADI campaign. These values of concentration are uncorrected. The figure also displays the surface friction velocity ($u_{*0}$) deduced by eddy covariance following Dupont et al. (2018), the air temperature ($T$) at 4 m height, and the relative humidity ($RH$) at 0.5 m height. During the campaign, the daily weather patterns were relatively constant, with clear skies and stronger northwesterly winds in the afternoon, combined with a peak in air temperature and a minimum in relative humidity. The most significant change occurred from September 23 to October 1 with a drier air, especially at night.

The dust concentration dataset exhibits a few gaps due to several factors: the Fidas was installed on September 13; the N3 had to be re-initialized on September 12; the pump of the Promo had to be replaced on September 20-21; and there was a power cut that affected both the Promo and the Fidas from late September 29 to early September 30. Despite being exposed to strong afternoon solar radiation, high air temperatures, and dusty winds, the N3 ran quite well during these 23 days.

The stronger afternoon winds produced almost daily dust emissions during a couple of hours. This is visible in Figure 3c-e as peaks in the dust concentration of particles larger than 0.8 $\mu$m (-0.1 in $\log_{10}$), which superpose on a slowly fluctuating background concentration. The N3 device captured remarkably well the same concentration dynamics as the two reference OPCs,

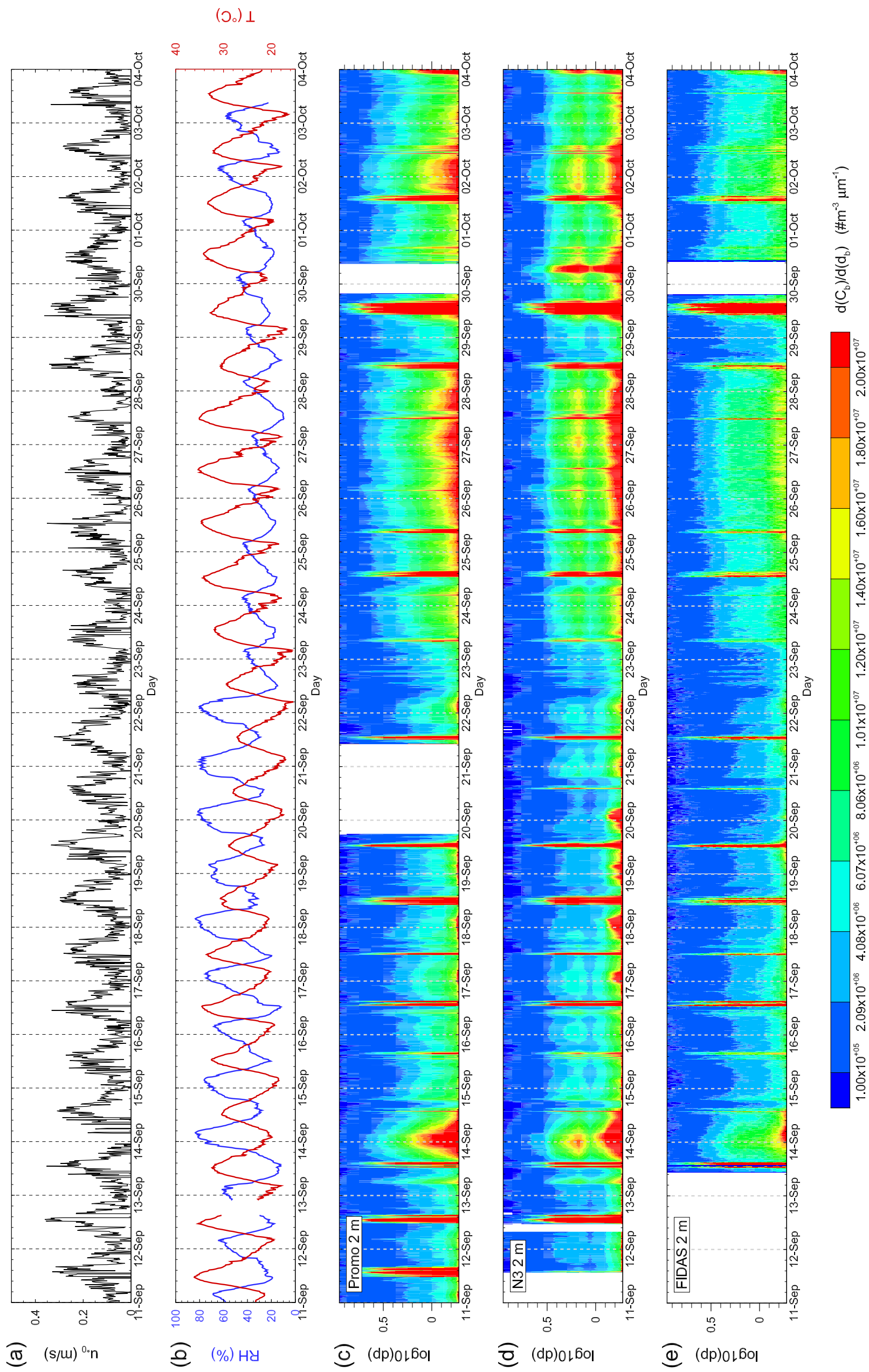

**Figure 3.** Time variations of the (a) surface friction velocity ($u_{*0}$) deduced from the sonic anemometers according to Dupont et al. (2018), (b) air temperature ($T$) at 4 m height and relative humidity ($RH$) at 0.5 m height, (c-e) mean size-resolved dust concentration in number recorded by the Promo, N3, and Fidas OPCs, respectively, around 2 m high and focusing on the particle size range of the Promo, 0.5 to 9 $\mu$m. The white areas correspond to periods without measurements.

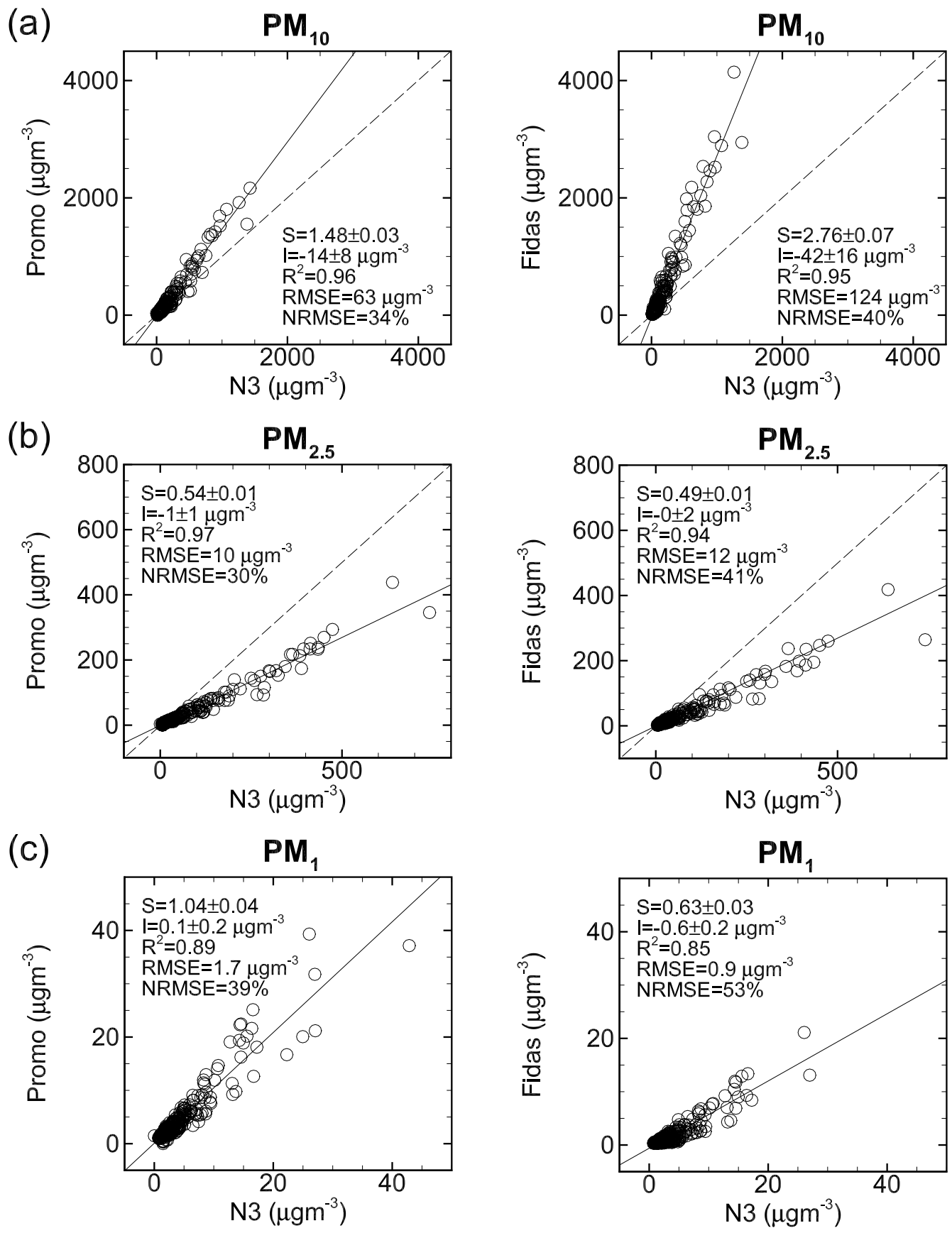

**Figure 4.** Comparison of the 15-min averaged $PM_{10}$ (a), $PM_{2.5}$ (b), and $PM_1$ (c) concentrations between N3 and Promo (left figures) and between N3 and Fidas (right figures), for periods with significant wind ($u_{*0} > 0.2\,\mathrm{m\,s^{-1}}$) during the J-WADI campaign. The solid lines represent the linear fits while the dashed lines represent the $1:1$ line. Performance of the linear fit are indicated in each plot with the slope (S), the intercept (I), the coefficient of determination ($R^2$), the root mean square error (RMSE), and the normalized RMSE (NRMSE).

including both the peaks in concentration related to dust events and the slower varying background particle concentrations. Visually, the magnitude of the N3 dust concentration seems close to the Promo concentration, with higher fine dust concentration ($< 0.8\,\mu$m) than the Fidas. However, the N3 concentration appears underestimated around the 1.0 $\mu$m size bin as compared to the two reference OPCs. This translates in Figure 3d into a suspicious line depicting lower concentration around 1 $\mu$m, which is not apparent in the data of the other OPCs. Conversely, the N3 overestimates the concentration of particles around 1.4 $\mu$m. This discrepancy around 1.0 $\mu$m was also observed for the seventeen N3 deployed in Iceland (result not shown). This suggests that the N3 may erroneously classify 1.0 $\mu$m particles into the larger size bin around 1.4 $\mu$m.

The evaluation of OPCs typically focuses on their capacity to estimate $PM_{10}$ or $PM_{2.5}$ concentrations. Without any correction, the 15-min average $PM_{10}$, $PM_{2.5}$ and $PM_1$ concentrations obtained by the N3 appear to be strongly correlated with those obtained from the two reference OPCs during the J-WADI campaign (Figure 4). Only 15-min periods with notable wind speeds ($u_{*0} > 0.2\,\mathrm{ms^{-1}}$), indicative of significant dust concentration, were considered in Figure 4. The performance of the N3 is slightly better when compared to the Promo because both OPCs were equipped with the same sampling head. The two OPCs exhibited high levels of correlation, with coefficients of determination $R^2$ exceeding 0.89 and normalised root mean square errors (NRMSE) below 39%. The N3 tends to underestimate coarse particles (lower $PM_{10}$) and overestimate particles between 1 and 2.5 $\mu$m (higher $PM_{2.5}$ while $PM_1$ are similar). These values of $R^2$ and NRMSE are consistent with those reported by Kaur and Kelly (2023) for $PM_{10}$ against the GRIMM aerosol spectrometer. This confirms the possibility of correcting the N3 concentration against one of the reference OPCs.

A notable erosion event occurred on September 29 (Figure 3), lasting for more than five hours. This event will be used hereafter to evaluate the N3 ability to estimate the dust flux in comparison to the Promo OPC (section 3.6). Prior to this, the N3 dust concentration will be corrected from a reference OPC (sections 3.2 and 3.3), and the fluctuations of the dust concentration measured by the N3 will be evaluated (section 3.4), which are essential for estimating the dust flux by eddy covariance.

### 3.2 N3 flow rate — First N3 concentration correction

Given that the N3 does not have a pump as the Promo, but a small fan aspirator, it is interesting to examine the temporal variation of the flow rate of this fan throughout the course of the J-WADI campaign (Figure 5a). Overall, the N3 flow rate $F_r$ was approximately 0.16 $\mathrm{Lmin^{-1}}$. This value is lower than the typical value mentioned by Alphsense, 0.28 $\mathrm{Lmin^{-1}}$, and can be attributed, at least in part, to the tube of the sampling head. Some notable decreases of the flow rate to as little as 0.05 $\mathrm{Lmin^{-1}}$ occurred during erosion events, but not consistently. For instance, $F_r$ does not exhibit a notable decline during the September 29th most intense event, in comparison to the majority of the other events. These variations in the N3 flow rate are in contrast to the near-constant 5 $\mathrm{Lmin^{-1}}$ flow rate of the Promo (Figure 5a). Note that for comparison purposes the same amplitude of variation of the flow rates around their mean values is used for both OPCs on the vertical axis of Figure 5a. This demonstrates that the absence of a pump in the N3 results in a more variable flow rate.

A more detailed examination of the N3 flow rate suggests a possible correlation between its decline and the wind direction during periods of high wind speed ($u_{*0} > 0.20\,\mathrm{ms^{-1}}$). Figure 5b shows a clear decline of $F_r$ as the wind direction shifted from 310° to 270°. This shift frequently happened during erosion events, except on September 29, where the wind direction remained approximately 300-310° (not shown). We suspect that this behavior is related to a pressure effect within the box where the N3 was located. This may have occurred specifically when the wind was blowing against the side of the box where an air outlet was located, around 270° (see Figure 2). This was confirmed by laboratory tests performed after the experiment. Installing a small exhaust system on the inlet or redesigning the box containing the N3 would remove this pressure effect and the associated flow rate decline.

This reduction of the N3 flow rate appears correlated with an overestimation of the concentration in small particles (less than 4 $\mu$m) by the N3. Indeed, the relative difference $\Delta C_{\mathrm{N3-Promo}}$ in concentration of small particles between the N3 and

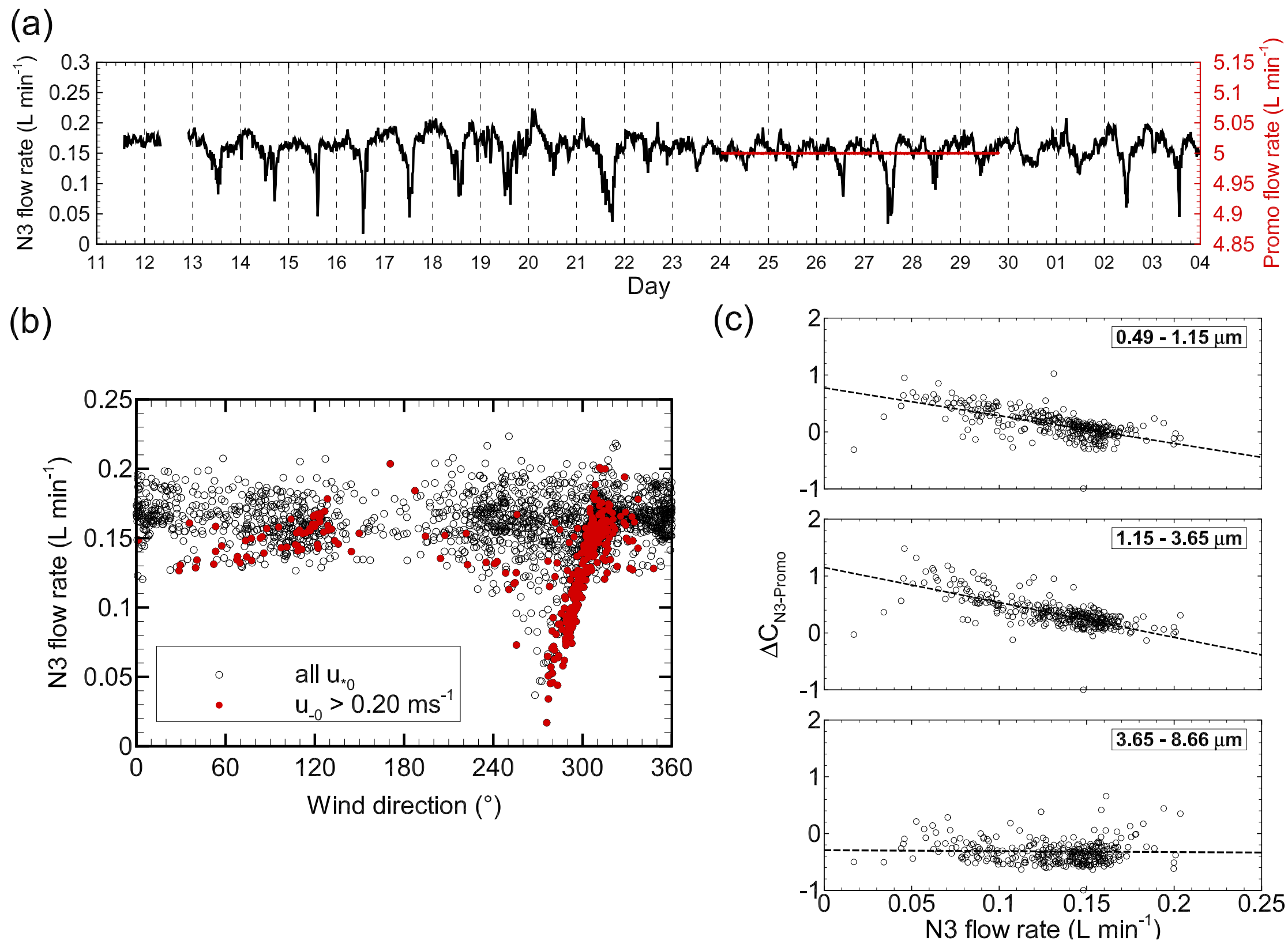

**Figure 5.** Evaluation of the N3 flow rate. (a) Time variations of the N3 flow rate (in black) during the 23 consecutive days of the J-WADI campaign, compared to the Promo flow rate (in red) during the 24 to 29 September period. (b) N3 flow rate as a function of the 2 m wind direction for all 15-min periods (black empty dots) and for 15-min periods where the friction velocity $u_{*0}$ exceeded $0.20\,\mathrm{ms}^{-1}$. (c) Relative difference $\Delta C_{\mathrm{N3-Promo}}$ in measured dust concentration between N3 and Promo, as a function of the N3 flow rate, for 15-min periods where $u_{*0} > 0.20\,\mathrm{ms}^{-1}$ and for three particle size ranges: 0.49-1.15 $\mu$m, 1.15-3.65 $\mu$m, and 3.65-8.66 $\mu$m. The dashed line represents a linear fit.

the Promo increases as the N3 flow rate declines (Figure 5c). Here, $\Delta C_{\mathrm{N3-Promo}} = \left(C_b^{\mathrm{N3}} - C_b^{\mathrm{Promo}}\right)/C_b^{\mathrm{Promo}}$, where $C_b^{\mathrm{N3}}$ and $C_b^{\mathrm{Promo}}$ are the dust concentrations of particle size bin $b$ measured by the N3 and Promo, respectively. For particles larger than 4 $\mu$m, the concentration may have been too low to observe a consistent effect of the decrease of the N3 flow rate on the measured concentration.

To correct this overestimation of the N3 dust concentration due to the flow rate reduction, we conducted a linear correlation analysis between $\Delta C_{\mathrm{N3-Promo}}$ and $F_r$ for each particle size bin over the entire campaign, focusing exclusively on 15-min periods where $u_{*0}$ exceeded $0.20\,\mathrm{ms}^{-1}$:

$$\Delta C_{\mathrm{N3-Promo}} = \Delta_0 + \Delta_F\left(F_r - \overline{F_r}\right), \tag{1}$$

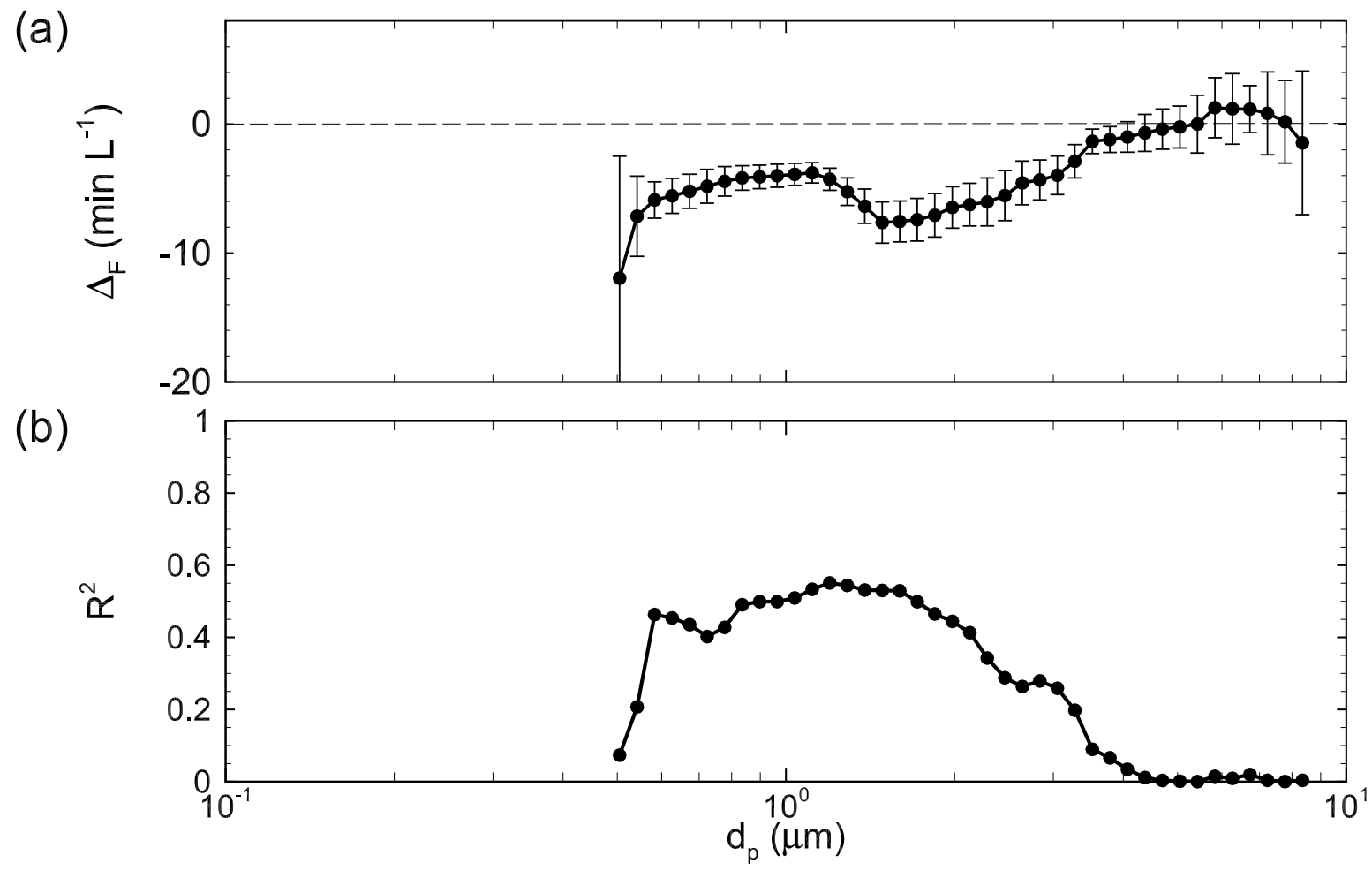

**Figure 6.** Results of the linear fit between the relative difference $\Delta C_{\mathrm{N3-Promo}}$ in dust concentration between the N3 and the Promo, and the N3 flow rate, as a function of the particle diameter $d_p$. $\Delta_F$ is the slope (a) and $R^2$ the coefficient of determination (b) of the linear fit analysis. Error bars represent the 95% confidence intervals.

where $\Delta_0$ is the intercept, $\Delta_F$ the slope, and $\overline{F_r}$ is the mean N3 flow rate over the campaign (0.16 Lmin$^{-1}$). For consistency, the size-resolved N3 concentration was interpolated to match the particle size bins of the Promo between 0.49 and 8.66 $\mu$m. Unlike $\Delta_F\left(F_r - \overline{F_r}\right)$, the intercept $\Delta_0$ represents the difference between the N3 and Promo concentrations not related to the flow rate variations. This latter term will be investigated in the following section. Here, we focus on the correction of the N3 concentration due to the flow rate variations (referred as the N3 first correction), i.e. the term $\Delta_F\left(F_r - \overline{F_r}\right)$.

For particles larger than 4 $\mu$m, the slope $\Delta_F$ is near zero (Figure 6a), meaning that the relative difference in concentration between the N3 and the Promo is not related to the flow rate variations. For smaller particles, $\Delta_F$ decreases with particles size, with a slight rise around 1.3 $\mu$m. The values of $\Delta_F$ were obtained with a reasonable coefficient of determination ($R^2 > 0.3$) between 0.6 and 2.5 $\mu$m (Figure 6b). To correct the N3 concentration, we used the obtained values of $\Delta_F$ for particle sizes between 0.54 and 5.42 $\mu$m, where the confidence intervals of $\Delta_F$ are small (error bars in Figure 6a). Above we considered $\Delta_F = 0$ minL$^{-1}$, and below $\Delta_F = -7$ minL$^{-1}$. This led to the following corrected N3 dust concentration:

$$C_{b,1}^{N3} = \left[1 + \Delta_F\left(F_r - \overline{F_r}\right)\right]^{-1} C_b^{N3}, \quad (2)$$

where the index ‘1’ in $C_{b,1}^{N3}$ refers to the N3 concentration after the first correction.

### 3.3 Calibration of the N3 dust concentration — Second N3 concentration correction

After the N3 first concentration correction, the remaining difference in concentration between the N3 and the Promo, corresponding to $\Delta_0$ in Eq. 1, is evaluated. For that purpose, we selected a calibration period corresponding to the five-day period prior to the September 29 erosion event, i.e. September 24-28. This period was chosen for the occurrence of significant dust

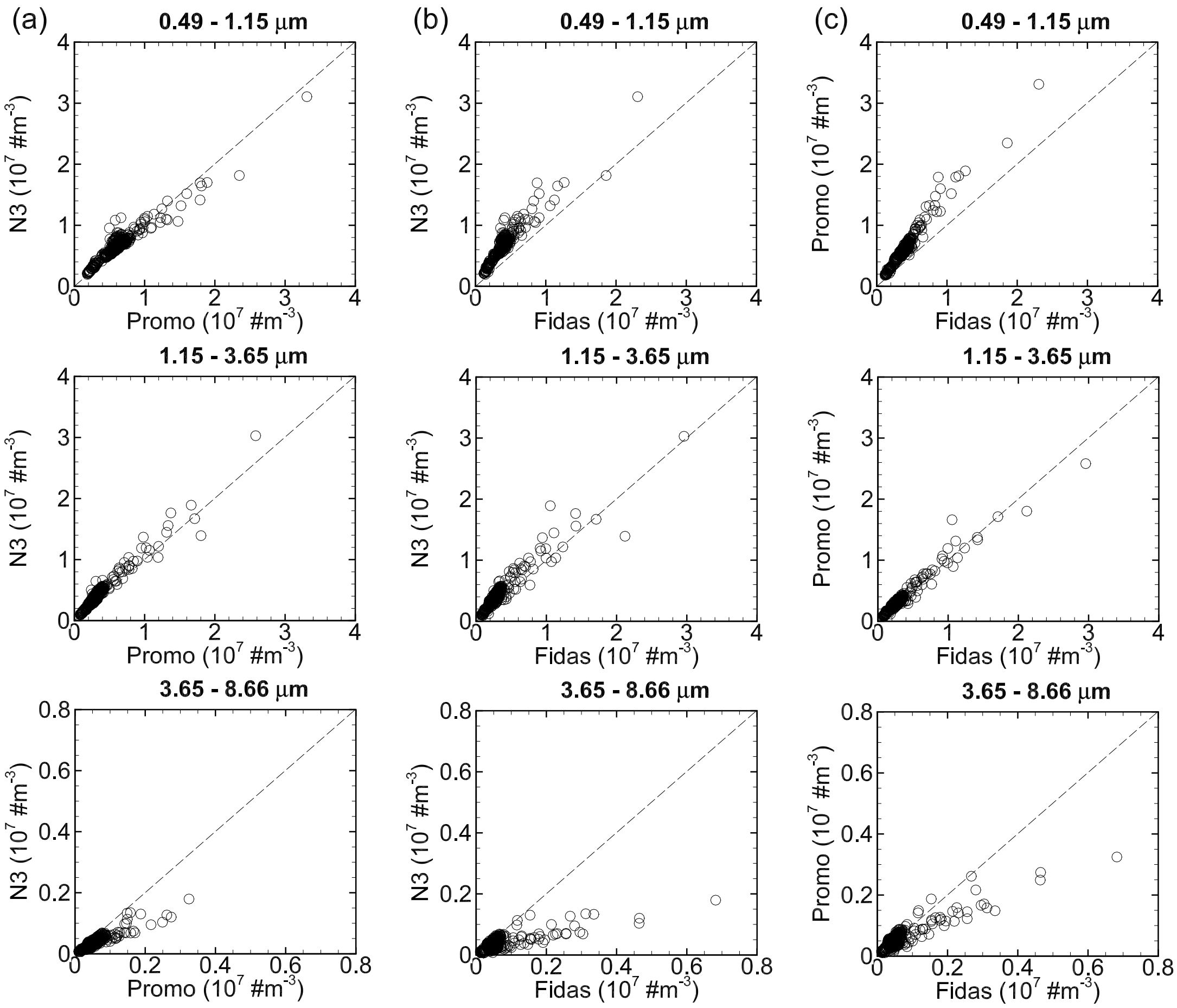

**Figure 7.** Comparison of the 15-min averaged dust concentrations between N3 and Promo (a), between N3 and Fidas (b), and between Promo and Fidas (c), during erosion events of the September 24th-28 period, for three particle size ranges: 0.49-1.15 $\mu$m, 1.15-3.65 $\mu$m, and 3.65-8.66 $\mu$m.

events and also because it followed the replacement of the Promo pump, which required recalibration of the instrument. The purpose of selecting a dust concentration calibration period that differs from the dust flux evaluation period (September 29) is to replicate the conditions of potential future experiments involving multiple low-cost OPCs and one reference OPC in a heterogeneous environment. The idea of such an experiment would be first, to estimate a calibration factor of the concentrations measured by the low-cost OPCs against the concentration of the reference OPC during an intercomparison period, then, to apply this calibration factor to the low-cost OPCs once deployed at different locations from the reference OPC in a heterogeneous environment.

The performance of the N3 in measuring dust concentration is first evaluated against the Promo during the September 24-28 period. As both OPCs used the same sampling head, conversely to the Fidas, any differences would only be related to

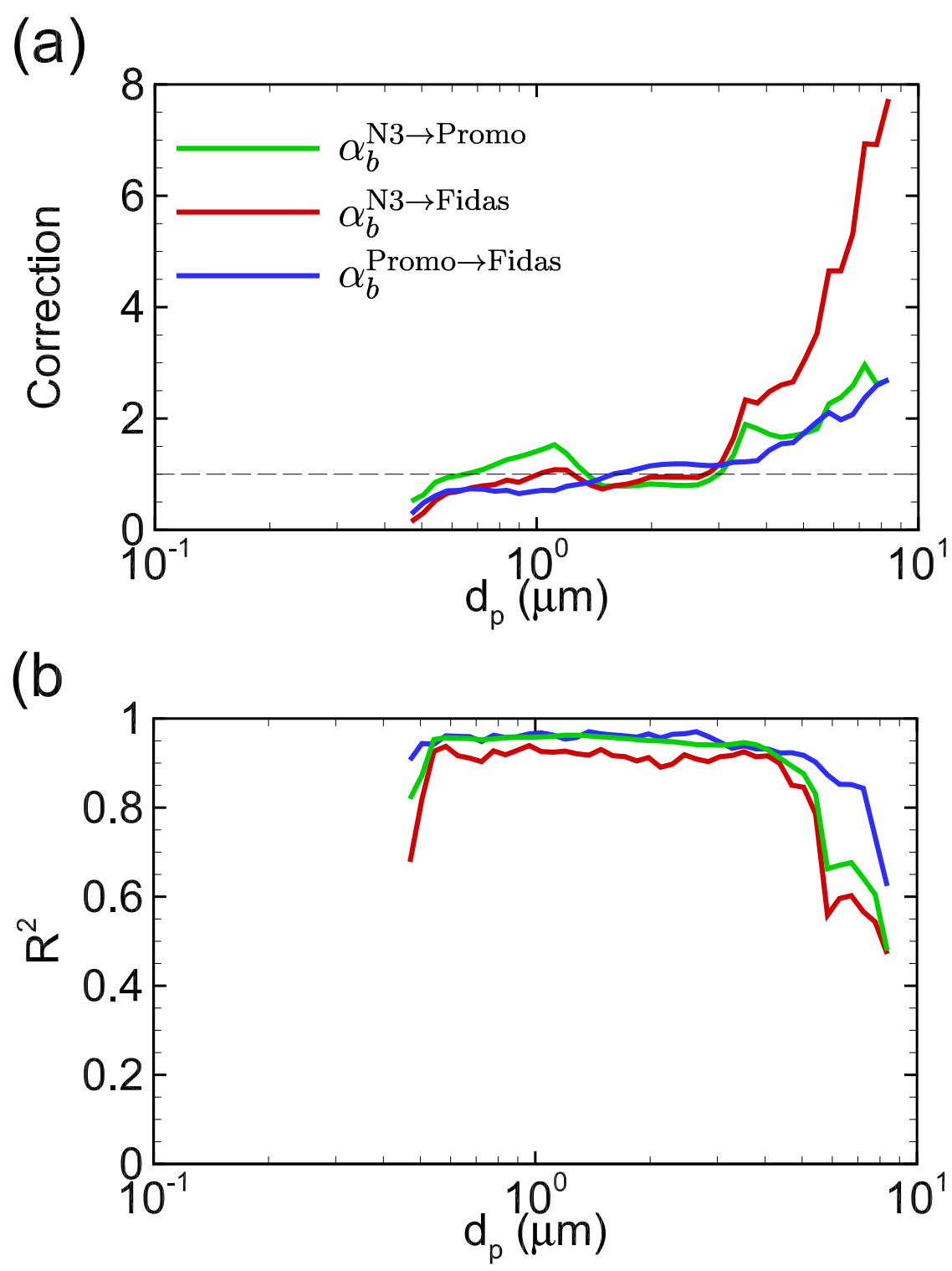

**Figure 8.** (a) Linear correction factors between N3 and Promo ($\alpha_b^{\mathrm{N3}\rightarrow\mathrm{Promo}}$), N3 and Fidas ($\alpha_b^{\mathrm{N3}\rightarrow\mathrm{Fidas}}$), and Promo and Fidas ($\alpha_b^{\mathrm{Promo}\rightarrow\mathrm{Fidas}}$), as deduced from the dust concentration intercomparison of the three OPCs during the September 24-28 period as a function of particle diameter $d_p$. (b) Coefficient of determination ($\mathrm{R}^2$) obtained from the intercomparison exercise. The horizontal black dashed lines indicate the perfect match values between OPCs: a correction factor equals to one.

differences between sensors themselves. Figure 7a presents in a scatter plot the 15-min average dust concentrations obtained from both the N3 and the Promo. To simplify the comparison, the concentrations are grouped into three size ranges common to the three OPCs (N3, Promo and Fidas): 0.49-1.15 $\mu$m, 1.15-3.65 $\mu$m, and 3.65-8.66 $\mu$m. For consistency, the N3 and Promo concentrations were interpolated to match the same size bin partition, the one corresponding to the Fidas. As observed for $\mathrm{PM}_{10}$, $\mathrm{PM}_{2.5}$ and $\mathrm{PM}_1$ concentrations (Figure 4), a distinct correlation exists between the N3 and Promo concentrations. The N3 appears to overestimate dust concentrations for particles smaller than 3.65 $\mu$m and to underestimate them for larger particles.

Calibrating the N3 dust concentration against the Promo concentration would lead to the calibration factors presented in Figure 8a as a function of the particle diameter. These calibration factors, denoted $\alpha_b^{\mathrm{N3}\rightarrow\mathrm{Promo}}$ (where the upper arrow points to the reference), have been calculated based on the linear correlation (slope value) between $C_{b,1}^{\mathrm{N3}}$ and $C_b^{\mathrm{Promo}}$. For particles smaller than 5 $\mu$m, the concentrations of the two OPCs correlate well with each other ($\mathrm{R}^2 > 0.7$, Figure 8b), enabling an accurate estimation of the calibration factor. For larger particles, the dust concentration was too low to achieve a good correlation

between OPCs. Over the entire particle size range, $\alpha_b^{\mathrm{N3}\rightarrow\mathrm{Promo}}$ varies from 0.5 to 3.0, i.e. the N3 concentration is about $+50\%$ to $-200\%$ of the Promo concentration. Around $1\,\mu$m, $\alpha_b^{\mathrm{N3}\rightarrow\mathrm{Promo}}$ reaches a peak value larger than one, followed by values lower than one for larger particles in the range between 1.5 and $2.5\,\mu$m. This behavior is consistent with Figure 3c-d, where we suspected an erroneous classification of $1\,\mu$m particles into the upper size bin.

As mentioned previously, the sampling head used by the N3 and Promo is well suited to perform eddy-covariance when the OPCs are collocated with a sonic anemometer. However, this sampling head is known to underestimate the concentration of coarse particles ($> 4\,\mu$m). The sampling head employed by the Fidas was more suitable for coarse particles, but its size was not adapted for eddy covariance. Moreover, the Fidas is expected to be less accurate for submicron particles as its sampling range started at 0.4 compared to $0.3\,\mu$m for the Promo. It would be possible to correct the N3 concentration against the Promo for submicron particles and against the Fidas for supermicron particles; however, this would introduce an element of confusion into this comparison exercise. For this reason, the decision was taken to calibrate the N3 and Promo dust concentrations against the Fidas concentration. The additional advantage of calibrating the N3 and Promo against the Fidas is to enable a comparison of the stability of the N3 calibration against that of the Promo for a period outside the calibration period.

The calibration factors of the N3 and Promo against the Fidas, $\alpha_b^{\mathrm{N3}\rightarrow\mathrm{Fidas}}$ and $\alpha_b^{\mathrm{Promo}\rightarrow\mathrm{Fidas}}$, respectively, were deduced using the same approach as $\alpha_b^{\mathrm{N3}\rightarrow\mathrm{Promo}}$. These calibration factors account for differences in concentration due to the sensors themselves and due to the sampling heads. The calibration factors were then extrapolated or interpolated to the initial size bins of the N3 and Promo. This led to the calibrated N3 and Promo dust concentrations, $C_{b,cal}^{\mathrm{N3}}$ and $C_{b,cal}^{\mathrm{Promo}}$, defined as follows:

$$C_{b,cal}^{\mathrm{N3}} = \alpha_b^{\mathrm{N3}\rightarrow\mathrm{Fidas}} C_{b,1}^{\mathrm{N3}} \tag{3}$$

$$C_{b,cal}^{\mathrm{Promo}} = \alpha_b^{\mathrm{Promo}\rightarrow\mathrm{Fidas}} C_b^{\mathrm{Promo}}, \tag{4}$$

where $C_{b,1}^{\mathrm{N3}}$ was already corrected for the N3 flow rate variability (Eq. 2).

Figure 8a presents $\alpha_b^{\mathrm{N3}\rightarrow\mathrm{Fidas}}$ and $\alpha_b^{\mathrm{Promo}\rightarrow\mathrm{Fidas}}$ as a function of the particle diameter. For particles smaller than 1.5-3 $\mu$m, $\alpha_b^{\mathrm{N3}\rightarrow\mathrm{Fidas}}$ exhibits the same behavior with particle size as $\alpha_b^{\mathrm{N3}\rightarrow\mathrm{Promo}}$ although $\alpha_b^{\mathrm{N3}\rightarrow\mathrm{Promo}}$ is larger. For larger particles, $\alpha_b^{\mathrm{N3}\rightarrow\mathrm{Fidas}}$ increases almost exponentially up to about 8. $\alpha_b^{\mathrm{Promo}\rightarrow\mathrm{Fidas}}$ increases almost linearly with particle size up to $4\,\mu$m, with values going from 0.5 to 1.5. Like $\alpha_b^{\mathrm{N3}\rightarrow\mathrm{Fidas}}$ for coarser particles, $\alpha_b^{\mathrm{Promo}\rightarrow\mathrm{Fidas}}$ increases exponentially with particle size. This last behavior should be related to the difference in sampling heads.

After calibration, the remaining discrepancy between the calibrated concentrations ($C_{b,cal}^{\mathrm{N3}}$ and $C_{b,cal}^{\mathrm{Promo}}$) and the Fidas concentration during the calibration period is roughly of the same order ($< 5\%$ in absolute value) between the N3 and the Promo (Table 1). Against the Promo, the N3 dust concentration is on average overestimated (6%) for the coarsest particle size range (3.65-8.66 $\mu$m), while for smaller particles the difference between both OPCs is low ($< 1\%$ in absolute value) (Table 1).

To evaluate $C_{b,cal}^{\mathrm{N3}}$ and $C_{b,cal}^{\mathrm{Promo}}$ outside the calibration period, their time variation is compared against $C_b^{\mathrm{Fidas}}$ in Figure 9a during the September 29 erosion event, for three particle size ranges: 0.49-1.15 $\mu$m, 1.15-3.65 $\mu$m, and 3.65-8.66 $\mu$m. Overall, the N3 accurately predicts the temporal dynamics of dust concentrations in comparison to the two reference OPCs. The difference does not exceed 19% in absolute value (27% in root mean square error) (Table 1). The Promo appears even in lower

**Table 1.** Comparison between calibrated dust concentrations obtained from the N3, Promo and Fidas OPCs, and between dust fluxes obtained from the N3 and Promo OPCs, for three particle size ranges: $0.49-1.15$ $\mu$m, $1.15-3.65$ $\mu$m, and $3.65-8.66$ $\mu$m, during the September 29 erosion event in Jordan. The comparison between calibrated dust concentrations is also presented during the calibration period (September 24-28). Comparisons are done through differences and Root Mean Square Errors (rmse) between dust concentration and fluxes in number. Both quantities are expressed in percentage of the dust concentration or flux of the last OPC. See Figures 9a and 14a for concentration and flux time variation during the erosion event, respectively.

| | 0.49 - 1.15 $\mu$m | | 1.15 - 3.65 $\mu$m | | 3.65 - 8.66 $\mu$m | |
|---|---|---|---|---|---|---|
| | diff. | rmse | diff. | rmse | diff. | rmse |
| *Dust concentration during the calibration period (September 24-28)* | | | | | | |
| N3 vs. Fidas | -1% | 18% | 1% | 26% | 1% | 44% |
| N3 vs. Promo | 1% | 14% | -0% | 18% | 6% | 32% |
| Promo vs. Fidas | -2% | 12% | 1% | 17% | -5% | 35% |
| *Dust concentration during the September 29 event* | | | | | | |
| N3 vs. Fidas | -6% | 12% | 5% | 12% | -14% | 26% |
| N3 vs. Promo | -19% | 27% | -5% | 9% | -10% | 17% |
| Promo vs. Fidas | 16% | 22% | 10% | 18% | -4% | 19% |
| *Dust flux during the September 29 event* | | | | | | |
| N3 vs. Promo | -14% | 125% | 21% | 105% | 26% | 100% |

agreement with the Fidas than the N3. Compared to the Promo, the N3 underestimates the dust concentration for all particle size ranges. The shape of the average size distribution of the dust concentration obtained from the N3 during the September 29 erosion event shows two main modes like the reference OPCs (Figure 9b), one around 0.7-0.8 $\mu$m and the other one around 1.5-2 $\mu$m. However, the N3 seems to underestimate the finest mode, which could be a consequence of the lower size bin resolution of the N3.

### 3.4 Concentration fluctuations

The concentration fluctuations measured by the N3 are evaluated against those obtained by the Promo during the September 29 erosion event. We compare first the main properties of the amplitude distribution of the fluctuations, commonly referred to as probability distribution function (PDF). Specifically, we analyzed the fluctuation standard deviation normalized by the mean concentration, and the fluctuation skewness and kurtosis, all estimated over 15-minute periods. Then, we compare the distribution of fluctuation energy according to the fluctuation time scales by looking at the fluctuation spectra. The Fidas was not included in this comparison because its sampling head introduced noise in high-frequency concentration fluctuations (not shown), rendering it unsuitable for performing eddy covariance.

Atmospheric Measurement Techniques
EGU
Discussions

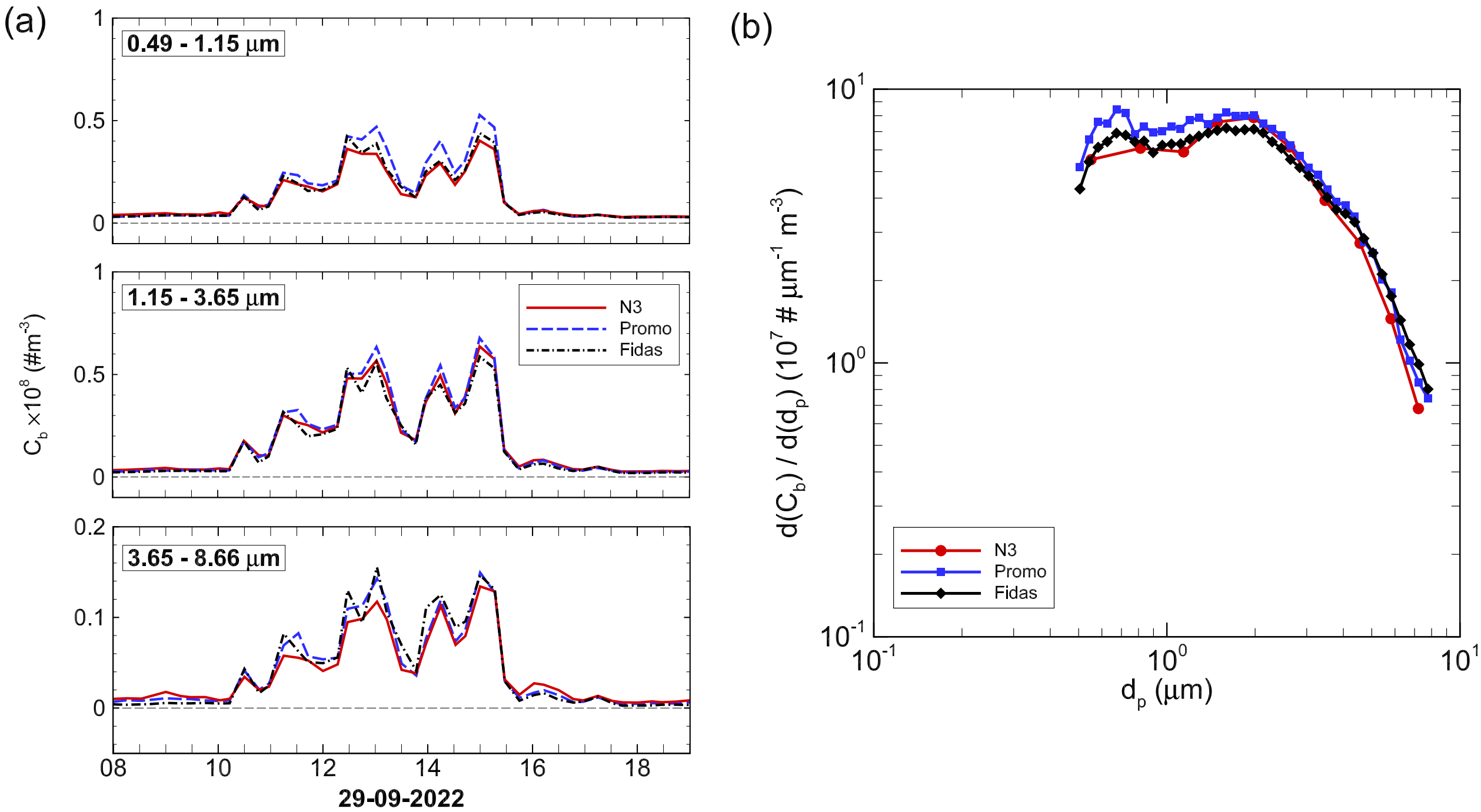

**Figure 9.** Comparison of the 2 m high dust concentration measured by the three OPCs during the September 29 erosion event, after calibration of N3 and Promo OPCs to the Fidas OPC during the September 24-28 period. (a) Time variation of the 15-min average dust concentration for three particle size ranges: 0.49-1.15 $\mu$m, 1.15-3.65 $\mu$m, and 3.65-8.66 $\mu$m. (b) Mean particle size distribution of dust concentrations during the erosion event. In (b) the concentration size distributions are following each OPC's own particle size discretization.

Figure 10 compares the time variation of the size-resolved standard deviation, skewness, and kurtosis of the dust concentration fluctuations obtained from the N3 and Promo. In general, the turbulent features of the N3 concentration seem comparable to those of the Promo, with similar dynamics and close amplitudes of statistical measures. During the erosion event, the fluctuations of submicron particles from the N3 seem less energetic (lower standard deviation) and less intermittent than those from the Promo.

In Figure 11a, the shapes of the ensemble-averaged 15-min energy spectra of dust concentration are compared between N3 and Promo, along with the shape of the 2-m high streamwise velocity $u$, vertical velocity $w$, and air temperature $\theta$ spectra. The spectra of meteorological variables exhibit the familiar shape of atmospheric surface-layer spectra as, for example, observed in Dupont et al. (2019) over a bare surface in Tunisia. Like meteorological variables, the dust spectra display a well-defined energy-containing range with a near $+1$ power law, up to about 0.03 Hz where $u$- and $\theta$-spectra reach their peak values. It is worth noting that at these low frequencies, the dust spectra exhibit fluctuations more comparable to those of the $u$-spectra than to the $\theta$-spectra. This suggests that the fluctuations of the streamwise wind velocity at these frequencies drive those of the dust concentrations, potentially initiating dust emission. At high frequencies, dust spectra are flatter, lacking a peak. For coarse

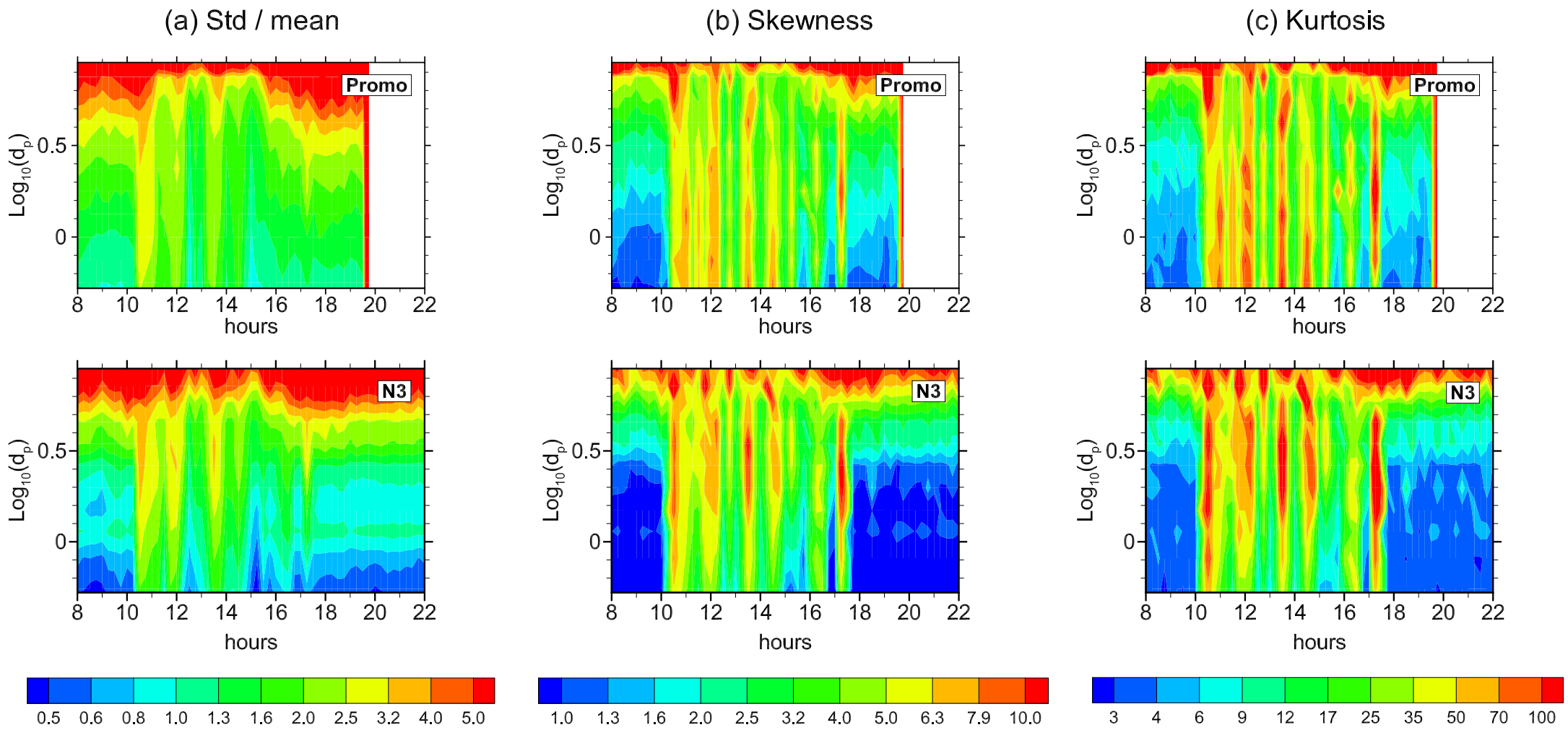

**Figure 10.** Comparison of the main turbulence characteristics of the dust concentration measured by the Promo (top) and N3 (bottom) OPCs during the September 29 erosion event as a function of time and particle size. These characteristics include (a) the standard deviation normalized by the mean concentration, (b) the skewness, and (c) the kurtosis, all calculated across 15-min periods. White areas correspond to periods without measurements.

particles, the spectra continue to increase with frequency, following a power law of $+1$, indicating white noise fluctuations dominance. The dust concentration was probably too low to identify a peak in the spectra as observed in Dupont et al. (2021). Both OPCs lead to dust concentration spectra with comparable amplitudes and shapes.

### 3.5 Variability of particle concentrations between N3 OPCs

The variability between the N3 OPCs in measuring particle concentrations is evaluated from the intercomparison performed in Iceland between seventeen N3 OPCs (see Section 2.1). Only a nighttime 16 min period with notable coarse particle concentration was observed during this intercomparison. Over this period, the flow rate of the N3 OPCs was on average $0.32 \pm 0.04\,\mathrm{L\,min^{-1}}$. All N3 exhibit similar dynamics of the 1-min average $PM_{10}$, $PM_{2.5}$ and $PM_1$ concentrations, albeit with some discrepancy in amplitude, reaching up to $\pm\,100\%$ of the mean concentration (Figure 12a-c). Furthermore, the mean number concentration particle size distributions during this 16-min period are similar between OPCs (Figure 12d), with a predominant mode around 10 $\mu$m and a significant proportion of submicron particles. These size distributions may be more representative of fog droplets than mineral dust [see, for example, Mazoyer et al. (2022) for fog droplets against Dupont et al. (2024) for mineral dust in Iceland]. Nonetheless, the type of particle is not relevant for the purposes of this intercomparison between N3 OPCs. Larger bias and NRMSE against the seventeen N3 mean concentrations are observed for particles within the one-micron range and for the coarsest particles (Figure 12e-f). Two N3 differ more from the others (brown and light-green lines in Figure 12). One of them had the lowest flow rate ($0.24\,\mathrm{L\,min^{-1}}$, brown line), while the flow rate of the other was close to the mean value

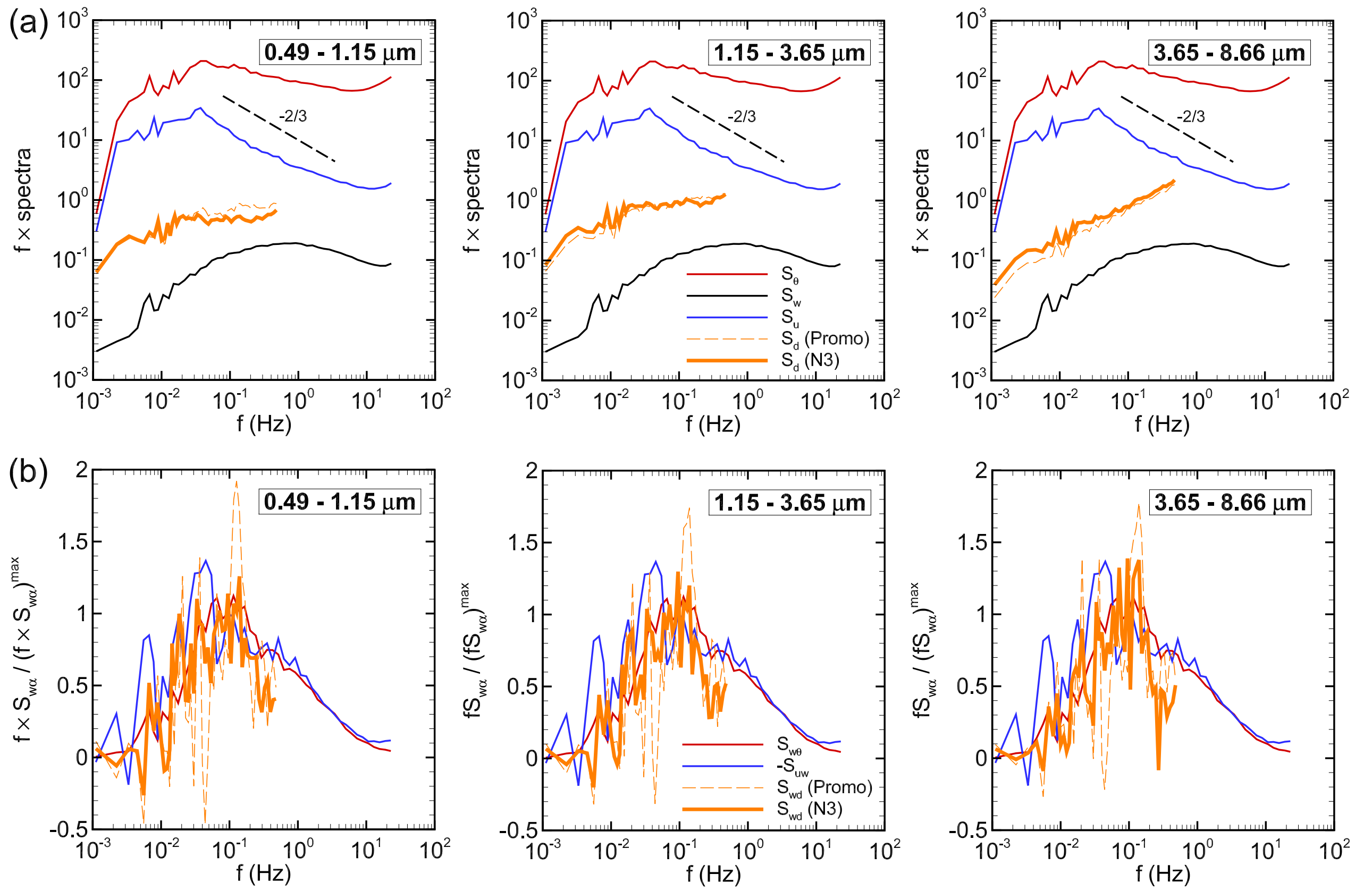

**Figure 11.** (a) Ensemble-averaged 15-min energy spectra of the longitudinal velocity ($S_u$), vertical velocity ($S_w$), air temperature ($S_\theta$), and dust number concentration ($S_d$) at 2-m height, for the September 29 event. Temperature, longitudinal velocity, and dust spectra are shifted upward to permit comparison. (b) Comparison of the ensemble-averaged 15-min normalized cospectra of the momentum ($S_{uw}$), heat ($S_{w\theta}$), and dust ($S_{wd}$) fluxes, at 2-m height, for the September 29 event. Cospectra are normalized by their maximum. The dust concentration spectra and dust-flux cospectra have been obtained for three particle size ranges (from left to right), and from both the Promo and N3 OPCs.

(0.34 Lmin$^{-1}$, light-green line). This evaluation of the concentration variability between N3 OPCs suggests that size-specific correction factors could be estimated for each N3.

### 3.6 Dust flux

Before comparing the dust fluxes obtained from the N3 and Promo, we first compare their dust-flux cospectra. The dust-flux cospectra allow to identify the time scale of the main turbulent structures contributing to the vertical dust fluxes. Figure 11b compares the ensemble-averaged 15-min dust-flux cospectra ($S_{wd}$) obtained at 2-m height from the N3 and Promo OPCs during the September 29 erosion event, for three particle size ranges. The figure also displays the momentum and heat flux cospectra ($-S_{uw}$ and $S_{w\theta}$) for the same event and height. The cospectra were normalized by their maximum values to facilitate the

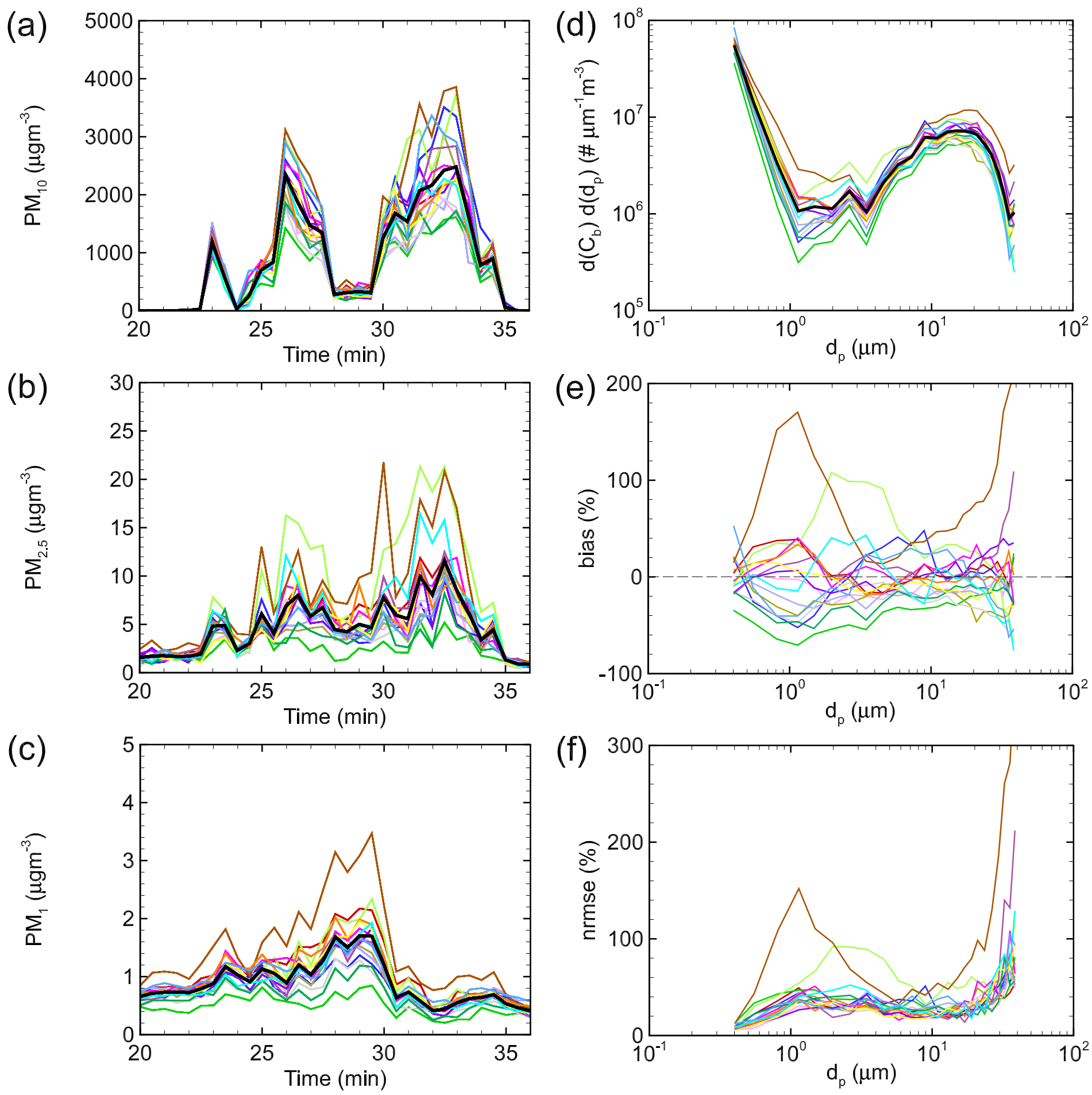

**Figure 12.** Variability of the measured particle concentrations between seventeen N3 OPCs over a 16 minute period. (a-c) Time variation of 1-min average $PM_{10}$, $PM_{2.5}$, and $PM_1$ concentrations recorded by the N3 OPCs. (d) Mean size distributions of the particle concentrations obtained from the seventeen OPCs. (e-f) Size-resolved biases and normalised root-mean-square-error (NRMSE) of the 1-min average concentrations measured by each N3 against the mean concentration of the seventeen N3 OPCs during the 16 minute period. Each thin line represents one N3 OPC, and the thick black lines in a-d represent the average concentration from the seventeen OPCs.

comparison of their distribution. Momentum- and heat-flux cospectra have similar shapes, although the moment-flux cospectra display greater fluctuations at low frequencies. The dust-flux cospectra obtained from the N3 exhibit similar shapes as those from the Promo, with even less fluctuations, suggesting that the high-frequency sampling of the N3 is as good as or even better, less perturbed, than that of the Promo. For both OPCs, the $S_{wd}$ cospectra present a peak close to the $S_{uw}$ and $S_{w\theta}$ cospectrum peaks, at approximately 0.1 Hz, and this for all particle size ranges. In the energy-containing range, the $S_{wd}$ and $S_{w\theta}$ cospectra have similar distribution, suggesting that dust is vertically transported or emitted by same large-scale motions as the heat. Passing the cospectrum peak, the $S_{wd}$ cospectra decrease on the high-frequency side as the $S_{uw}$ and $S_{w\theta}$ cospectra,

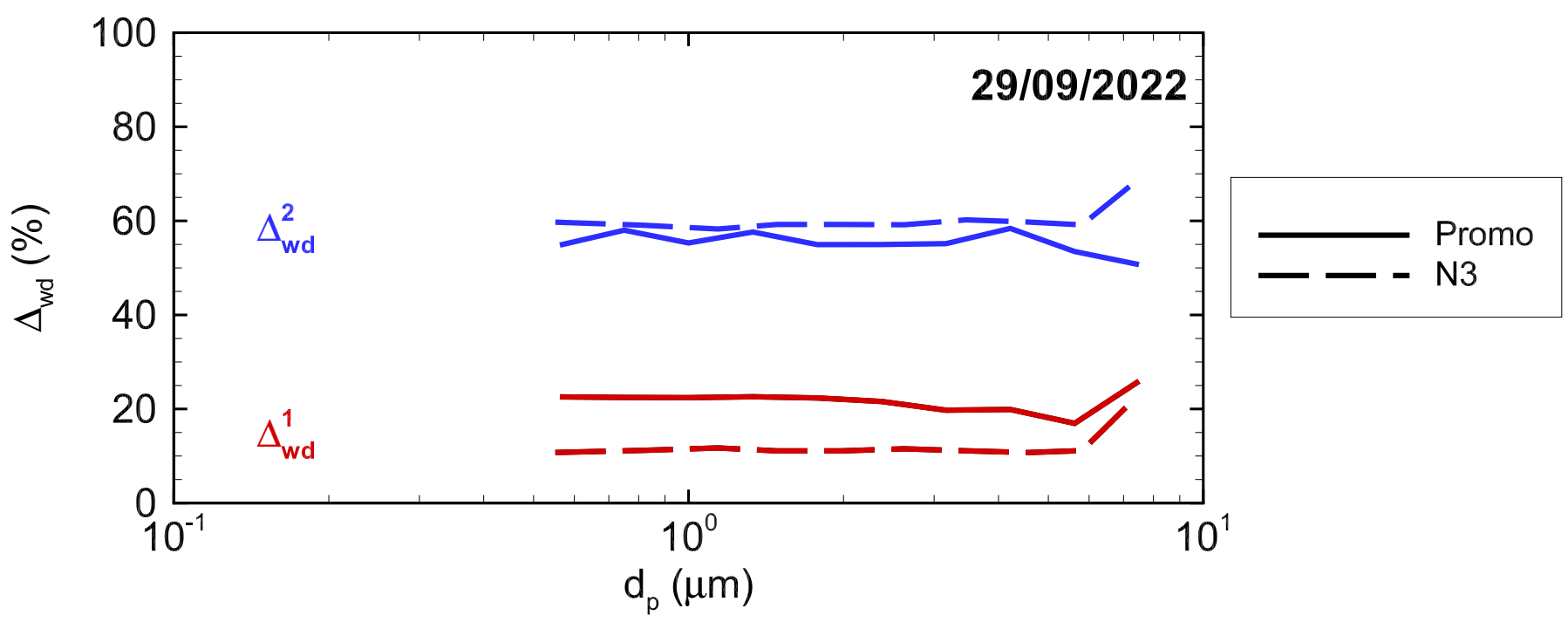

**Figure 13.** Variation of the high-frequency corrections of the dust flux as a function of the particle size, when corrections are either based on a standard cospectrum shape fitted on the obtained dust flux cospectrum ($\Delta^1_{wd}$) or on comparing the dust flux cospectrum with the momentum flux one ($\Delta^2_{wd}$), for the September 29 erosion event, and for the 2-m high Promo and N3 OPCs.

but with a steeper slope in the inertial subrange, indicating a possible loss of the dust flux at high frequency due to the limited high-frequency of the OPCs (1 Hz).

The missing dust flux at high frequencies has been assessed using two approaches outlined in Dupont et al. (2024). The first method involves comparing the dust-flux cospectra with the typical shape of scalar flux cospectrum, while considering an attenuation on the high-frequency side due to the slow response of the OPC. The second method estimates the high-frequency losses of the dust-flux cospectra based on momentum-flux cospectra, assuming similar transport mechanisms for dust and momentum at high frequencies. Figure 13 illustrates the high-frequency losses of dust flux, denoted as $\Delta^1_{wd}$ and $\Delta^2_{wd}$, respectively, for the September 29 erosion event as a function of the particle size. On average, $\Delta^1_{wd}$ represents about +20% of the dust flux for the Promo and only +10% for the N3, with no evident pattern with particle size for both OPCs as observed in Dupont et al. (2021). The second method yields a higher correction compared the first method. On average, $\Delta^2_{wd}$ represents between +50% and +55% of the dust flux for the Promo and N3, respectively, with no clear trend with particle size as $\Delta^1_{wd}$. This higher correction can be attributed to the larger contribution of low frequencies to the momentum-flux than to the dust-flux cospectra. A correction based on the heat flux cospectra would have been lower.

Figure 14a compares the EC dust fluxes obtained from the N3 and Promo during the September 29 erosion event. Both fluxes have been corrected for high-frequency losses using the first method correction ($\Delta^1_{wd}$) as done in (Dupont et al., 2021). Overall, the dust flux obtained from the N3 is comparable to that obtained from the Promo, with a difference of less than 26% (125% in rmse) (Table 1). The size distribution of the dust flux is relatively well captured as well by the N3, with a more pronounced dominant mode around 2 $\mu$m (Figure 14b). The N3 dust flux is lower for particles smaller than 1.15 $\mu$m, and higher above. The N3 flux displays slightly less fluctuations than the Promo, with less periods without dust flux during the event (Figure 14a). We suspect that the N3 acquires better the 1 Hz fluctuations of dust concentration than the Promo, as suggested by the less

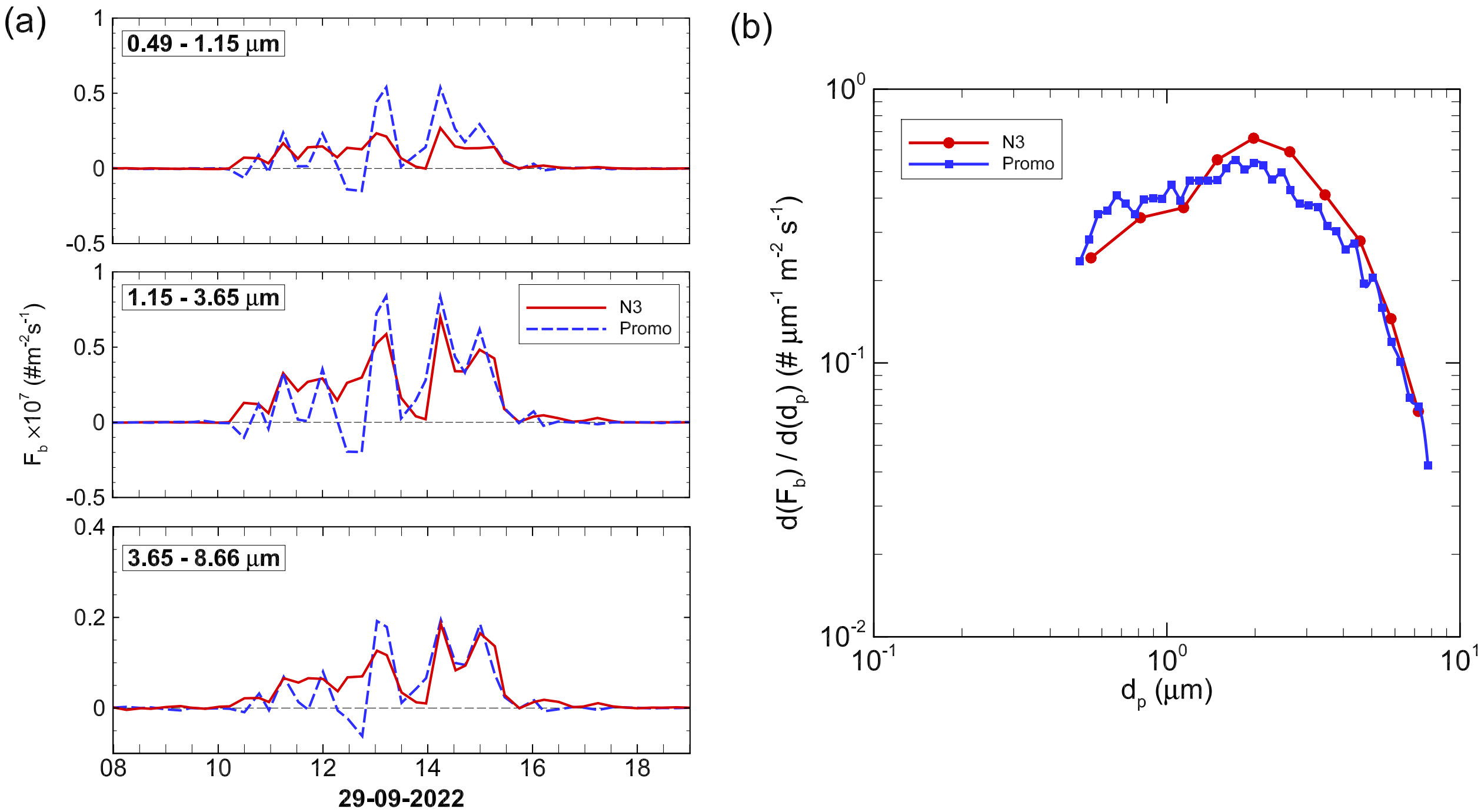

**Figure 14.** Comparison of the dust flux obtained from the N3 and Promo OPCs during the September 29 erosion event, after calibration of N3 and Promo concentrations to the Fidas one during the September 24-28 period. (a) Time variation of the 15-min average dust flux for three particle size ranges: 0.49-1.15 $\mu$m, 1.15-3.65 $\mu$m, and 3.65-8.66 $\mu$m. (b) Mean particle size distribution of dust flux during the erosion event. The Promo and N3 dust fluxes have been deduced using the eddy covariance method. In (b) the dust flux size distributions are discretized following each OPC own particle size resolution.

fluctuating N3-dust-flux cospectra (Figure 11b). This implies a more continuous correlation between the dust concentration and vertical velocity fluctuations, and thus a more continuous dust flux during the event for the N3.

## 4 Conclusions

The low-cost OPC-N3 (Alphasense) has been evaluated for its ability to estimate size-resolved dust fluxes during aeolian soil erosion events when combined with a sonic anemometer, using the eddy-covariance method. This evaluation involved a comparison with the reference OPCs Promo 2000, focusing on particle diameters ranging from 0.5 to 9 $\mu$m.

Overall, the low-cost OPC was able to measure surprisingly well the particle concentration over 23 consecutive days in the Wadi Rum desert of Jordan, despite the challenging environmental conditions, including dusty winds and warm temperatures. Two main discrepancies were identified. Firstly, the N3 flow rate presented fluctuations over time, which resulted in an overestimation of the concentration of fine dust particles as the N3 flow rate decreased. Fluctuations in the flow rate of the reference OPC were not observed, the pump of this latter OPC ensuring a more stable flow rate. Secondly, some of the particles near

1 $\mu$m in diameter have been classified by the N3 into the upper size bin, resulting in an underestimation of the 1 $\mu$m size bin and an overestimation of the bin above it. After correction for both discrepancies, the predicted N3 concentration was found as good as that of the Promo compared to a third OPC (Fidas 200S). Furthermore, the low-cost OPC yielded a similar concentration particle size distribution as the reference OPC. The turbulence characteristics of the dust concentration measured by the N3, namely the concentration high-frequency fluctuations, were also comparable to those of the reference OPC in terms of amplitude distribution and energy density spectrum. Between several N3 OPCs, the variability in measured particle concentrations can be significant, with a potential range of up to $\pm 100\%$. However, the high correlation between OPCs suggests that this variability can be mitigated by the application of correction factors deduced during an intercomparison exercise from size-specific linear regression.

Unlike some previous studies, the performance of the N3 OPC during the J-WADI campaign appeared unaffected by humidity conditions, but this may be explained by the similar meteorological conditions observed during dust emission events, with a relative humidity ranging from approximately 20 to 30%.

The dust fluxes obtained from the N3, by correlating the dust concentration fluctuations with vertical wind velocity fluctuations, were consistent with those obtained from the reference Promo OPC. This leads to close size distributions of the emission dust flux. Differences were in the range of $\pm 30\%$.

This study demonstrated the ability of the OPC-N3 to estimate size-resolved dust fluxes via the eddy-covariance method, after calibrating the dust concentration from a reference OPC. Due to their lower cost, compact size, and low power consumption, these devices appear promising for expanding the spatial resolution of dust emission in complex environments.

*Data availability.* The processed data used in this study will be available once the paper has been accepted for publication.

*Author contributions.* SD drafted the manuscript with input from EL, KK, MK, and CPG-P. SD analyzed the data and prepared all figures. MI and JB designed OPC-N3 acquisition and setup. KK conducted the OPC-N3 intercomparison in Iceland. AA, AGR, CGF, CPG-P, KK, MK, MRI, SD and XQ implemented the field campaign. CPG-P and MK proposed and designed the J-WADI campaign with contributions from AA, KK, SD and XQ.

*Competing interests.* The authors declare that they have no conflict of interest.

*Acknowledgements.* We acknowledge the European Research Council under the Horizon 2020 research and innovation programme through the ERC Consolidator Grant FRAGMENT (grant agreement no. 773051), and the AXA Research Fund through the AXA Chair on Sand and Dust Storms at BSC for financial support of the field campaigns. We also thank the Helmholtz Association's Initiative and Networking Fund (grant agreement no. VH-NG-1533) for financial support of the field campaign in Jordan. S. Dupont acknowledges the financial support

of the Department Agroecosystem of INRAE. K. Kandler is funded by the Deutsche Forschungsgemeinschaft (DFG, German Research Foundation) – 416816480; 417012665. Finally, we acknowledge efforts by the staff at Wadi Rum Protected Area, Aqaba Special Economic Zone Authority, and Directorate of Environmental Monitoring and Assessment at Ministry of Environment.

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
