# Peer review of "Performance of a low-cost optical particle counter (Alphasense OPC-N3) for estimating size-resolved dust emission flux using eddy covariance"

_Atmospheric Measurement Techniques, 2024_

## Author Comment (AC1)

**Response to the referee 1**

We thank the referee for his/her critical assessment of our work and his/her interesting suggestions that contributed to improve our paper. In the following we address their concerns point by point.

Low-cost sensors can be an affordable and versatile tool to improve our understanding of the air quality in more diverse scenarios. Their shortcomings which mostly come from their low-cost nature and are not currently fully understood, are the main limiting factor for their wider application as a sensible alternative or supplement to the existing network. Thus, studies as this one can help in improving our understanding on the LCS and extend their use in scenarios where they can provide useful information. The study presented in the manuscript is done in a concise and scientifically correct way. I have no major concerns about the work presented and I can only suggest few things which may improve its readability and content.

To start with, my main concern about the paper is its size. I think that the paper is way too long. While I did not find many things that were unnecessary, there is too much detail on everything. I would suggest the authors to try and reduce the size of the chapters as much as possible and add a take-away message at the end of each chapter as the lack of a discussion or a sum-up chapter is not helping in having clear conclusions of the analysis done in each chapter (which I do not suggest with the present form of the paper as it would further increase its size).

Reply: We understand the concern regarding the length of the paper and the level of detail presented. Our aim was to provide a comprehensive and transparent account of the methodology and findings to ensure clarity for the readers. We believe that the paper's length is consistent with the average for AMT papers.
Nevertheless, we have carefully revisited the manuscript to identify areas where the text could be streamlined without losing essential information. After thorough review, we found only minor edits that could be made to reduce the text. However, some revisions requested by the reviewers have led to a slight increase in length, including the addition of an extra figure.
We acknowledge the importance of clear conclusions. However, we believe that including a take-away message at the end of each subsection would significantly increase the paper's length, particularly in the "Results and Discussion" section, which has six subsections. Instead, we have ensured that the main messages are clearly summarized in the Conclusion section, maintaining a concise and cohesive narrative.

Adding to that the abstract is poorly written in my opinion. The study has a multitude of results and while many of these point the good performance the OPC-N3 had, this is briefly demonstrated in two rather vague sentences. As the abstract is probably the first part everyone reads, I think it should include more results (even in the form of simple conclusions coming from the analysis done, mentioning what was tested), which would make the reader more interested to carry on reading the paper.

Reply: The abstract has been revised to provide more detailed information on (1) the turbulence characteristics of dust fluctuations explored, including variance, skewness, kurtosis, and the energy spectrum, and (2) the main discrepancies observed with the N3, specifically flow rate fluctuations and the misclassification of particles around $1\,\mu$m in the upper bin, which were subsequently corrected.

In the Methodology chapter 2.2 while there is a detailed description of the three OPCs there is no

information about their price range, even as a ratio of one against the other. Adding to that, there is no clear information on what makes the N3 cheaper and how these differences are expected to affect the quality of the data collected, as the way it is presented it appears as the N3 is the same as the others but consuming less energy and having no pump.

**Reply**: The price range of the OPCs was given in the introduction section: "*low-cost refers to sensors costing around 50 to 1000€ compared to 20-25 k€ for traditional OPCs.*" and "*the Alphasense OPC-N3 ($\approx 600$€)*".
The lack of detailed information provided by constructors regarding their OPCs complicates the ability to fully explain the price differences among them. The lower price of the N3 is likely due to (1) the absence of an acquisition system and a software for data plotting and statistical calculations, (2) the absence of a flow control and a sheath-flow system for stabilizing the aerosol flow prior to entering the optical chamber, (3) larger production tolerances, (4) the use of less expensive materials, optical components, etc.
This is now specified in section 2.2.

The N3 is known to be greatly affected by humid conditions. While it is mentioned that what is studied in this paper is its behaviour during dust events (happening during the daytime), looking at figure 3 this can possibly be the case for the nights of the 14,17-22/9, when RH reached 80%. Was there a notable discrepancy in these periods? It would be a good opportunity to look at this and provide valuable information for one of the main problems the N3 has. I see that this is slightly discussed in the conclusions but only mentions the periods with low RH.

**Reply**: Figure 1b below compares the 15-min averaged $PM_{10}$, $PM_{2.5}$, and $PM_1$ concentrations between the N3 and the Promo for the humid night of 13-14 September ($RH > 70\%$). We preferred to focus on this night because the others had a too low concentration of particles (see Figure 3 in the paper). Under these more humid conditions, the performance of the N3 does not appear to be diminished when compared to that observed during the drier conditions of wind erosion events (Figure 1a). Figure 1b is now included in the revised manuscript, and our conclusions (section 4) on the impact of humidity has been updated accordingly:
"*Unlike some previous studies, the performance of the N3 OPC during the J-WADI campaign appeared unaffected by humidity conditions. This can be attributed to the similar meteorological conditions observed during dust emission events, with a relative humidity ranging from approximately 20 to 30%. The comparison of the N3 with the Promo on a humid night ($RH > 70\%$) did not reveal any performance degradation of the N3 relative to drier periods. Unlike the Fidas, the fact that the N3 and the Promo do not dry the sampled air, renders both OPCs sensitive to water droplets, potentially explaining why their recorded concentration differences appear unaffected by air humidity. However, measurements in prolonged humid conditions are required to confirm our observation.*"

[Figure]

Figure 1: Comparison of the 15-min averaged $PM_{10}$ (top), $PM_{2.5}$ (middle), and $PM_1$ (bottom) concentrations between N3 and Promo and between N3 and Fidas, for (a) periods with significant wind ($u_{*0} > 0.2\,\mathrm{ms}^{-1}$) during the J-WADI campaign, and (b) for the humid night of 13-14 September (RH > 70%). In humid conditions (b), the Fidas results are not comparable to those of N3 and Promo due to the fact that the Fidas dries the sample air. The solid lines represent the linear fits while the dashed lines represent the 1 : 1 line. Performance of the linear fit are indicated in each plot with the slope (S), the intercept (I), the coefficient of determination ($R^2$), the root mean square error (RMSE), and the normalized RMSE (NRMSE).

Throughout the analyses done it is noted that the N3 and Promo seem to perform similarly (in good or bad cases) compared to the FIDAS. One of my guesses is that the positioning and the sampling heads are the factors that affected this outcome. Though I do not suggest the removal of the FIDAS information the question remains. If the Promo was used as the reference for the N3, what is the purpose of the FIDAS, as it makes the Promo appear to underperform as well? I suggest that you add a clear note in the results that while the FIDAS is a reference instrument its measurements are probably biased by these factors, should be considered as "background measurements" and direct comparisons should be evaluated cautiously.

**Reply**: Our choice to use the FIDAS as a reference is explained as follows in section 3.3:
"*As mentioned previously, the sampling head used by the N3 and Promo is well suited to perform eddy-covariance when the OPCs are collocated with a sonic anemometer. However, this sampling head is known to underestimate the concentration of coarse particles ($> 4\,\mu m$). The sampling head employed by the Fidas was more suitable for coarse particles, but its size was not adapted for eddy covariance. Moreover, the Fidas is expected to be less accurate for submicron particles as its sampling range started at 0.4 compared to $0.3\,\mu m$ for the Promo. It would be possible to correct the N3 concentration against the Promo for submicron particles and against the Fidas for supermicron particles; however, this would introduce an element of confusion into this comparison exercise. For this reason, the decision was taken to calibrate the N3 and Promo dust concentrations against the Fidas concentration. The additional advantage of calibrating the N3 and Promo against the Fidas is to enable a comparison of the stability of the N3 calibration against that of the Promo for a period outside the calibration period.*"
In the same section, when evaluating the calibration factors, it is now clarified that "*the calibration factors of the N3 and Promo against the Fidas, [...] account for differences in concentration due to the sensors themselves, to the sampling heads, and to the difference of location between the N3 and Promo and the Fidas*".

Minor additions suggested

Introduction, line 21: I would suggest adding the chemical composition of the particles on their environmental impact, as it is one of the most important factors both in their direct impact as well as for their evolution.

**Reply**: Done.

Results, line 131: It is pointed that the N3 needed a restart. What was the reason for that? Stability is one of the factors affecting the reliability of low-cost sensors and any information for this is useful for future users.

**Reply**: This restart was related to our acquisition system and not to the N3 itself. This has been clarified in the text.

Was response time looked at? The reaction time of the instruments on fast changes is important when looking at specific conditions. This would be interesting information if available.

**Reply**: To illustrate the N3 response time, the high-frequency (1 Hz) $PM_{10}$ fluctuations recorded by the N3 is compared to those of the Promo in Figure 2 below for the initial hour of the September 29 erosion event. Despite the high intermittency of the dust concentration, the time response of the N3 aligns closely with that of the Promo. The timing of the main peaks of dust concentration recorded by the Promo seems well captured by the N3. This new figure is now included in the manuscript.

[Figure]

Figure 2: Time series of the high-frequency (1 Hz) $PM_{10}$ measured by the N3 and the Promo, without any correction, during the initial hour of the September 29 erosion event.

Note that the main properties of the amplitude distribution of the particle concentration fluctuations, that are the size-resolved standard deviation, skewness, and kurtosis, as well as the distribution of fluctuation energy according to the fluctuation time scales, are discussed in section 3.4.

**References**

---

## Author Comment (AC2)

**Response to the referee 2**

We thank Amato Evan for his critical assessment of our work and his interesting suggestions that contributed to improve our paper. In the following we address their concerns point by point.

This paper presented results from an observational study designed to evaluate the feasibility of using a so-called low-cost optical particle counter (OPC) to generate high frequency measurements of the size resolved dust concentration, with the specific application to using the eddy covariance method to estimate the vertical turbulent diffusive dust flux. Via comparison with reference instruments, the authors determined that the vertical flux estimated with the low-cost instrument was comparable to that from a (more expensive) reference instrument. I found this manuscript to be thorough in terms of evaluating/comparing the N3's characteristics with the reference OPCs and only have a three comments that I would like to see addressed before I can recommend the manuscript for publication in AMT.

Comments:

Section 2. Can the authors report on the wavelength of light that the three OPCs operate at and how differences in this characteristic may influence the calculated size-resolved particle concentration? I'm specifically thinking about the influence of the (spectrally resolved) particle complex refractive index on the estimated particle sizes (e.g., Huang et al. 2021: Linking the different diameter types of aspherical desert dust indicates that models underestimate coarse dust emission, GRL)

**Reply**:  The N3 utilizes a laser light source, whereas the Promo operates with a xenon source, and the Fidas employs an LED source. The manufacturer does not provide the light source wavelength for the Promo and Fidas. The N3's wavelength ranges from 600 to 650 nm according to *Kaur and Kelly* (2023).
Each of the three OPCs has undergone calibration with monodisperse, non-absorbing polystyrene latex spheres (PSLs). The refractive index is $1.50 + 0i$ for the N3 (default value) and $1.59 + 0i$ for the Promo and Fidas (latex value). The Promo and Fidas underwent calibration at the start of the experiment, whereas the N3 comes pre-calibrated from the factory. Hence, the three OPCs consider spherical particles with size corresponding to optical diameters, and provide particle size distributions in terms of PSL-equivalent diameters, producing the same scattered light intensity as the measured (absorbing and irregular) dust particles, but at a different wavelength range depending on the OPC.
Since the three OPCs have been calibrated using the same particles, we assume that the calibrations of the N3 and Promo relative to the Fidas, as described in section 3.3, inherently account for differences in optical diameters due to different wavelengths of each instrument, along with other factors like the inlet design. Given this calibration, we propose that the conversion of N3 particle size distributions (PSDs) to dust geometric diameters, following *Huang et al.* (2021), may use the same conversion as the reference instrument, *i.e.,* the Fidas.
This discussion has been incorporated into the manuscript.

Section 3.1 Can/should the apparent tendency of the N3 OPC to misclassify 1 um particles into the 1.4 um bin be accounted for by summing the particle counts for these two bins? This could also help to address the related bias shown in Fig 14b at these same sizes.

**Reply**:  This may indeed serve as a solution; however, it would diminish the size resolution of the N3 dust concentration, which is already coarse in comparison to the Promo within this size range. Regarding

this comment, this tendency of the N3 OPC to misclassify $1\,\mu$m particles into the $1.4\,\mu$m bin should not affect the $PM_{2.5}$ and $PM_{10}$ estimations.

Section 3.2 I am afraid I don't understand why the reduction in the N3 flow rate results in an overestimation of the N3 concentration relative to the Promo, given that the N3 flow rate is accounted for in the concentration calculation (line 91). Can the authors provide an explanation why accounting for the flow rate in the concentration calculation is nonetheless insufficient to account for the wind direction bias?

**Reply**: This is a good point for which we do not have an explanation. The only issue we anticipated with the decrease in the N3 flow rate is that air sampled at 1 Hz may be less representative of the ambient air, potentially causing problems for low concentrations, probably underestimating the concentration but not overestimating it. The correlation between the reduction of the N3 flow rate and the overestimation of particle concentration may suggest either an underestimation of the flow rate by the sensor or the existence of an unknown calibration factor somewhere hard-coded in the N3, and linked to the N3 average flow rate.

**References**

Huang, Y., A. A. Adebiyi, P. Formenti, and J. F. Kok, Linking the different diameter types of aspherical desert dust indicates that models underestimate coarse dust emission, *Geophysical Research Letters*, *48*(6), doi:10.1029/2020gl092054, 2021.

Kaur, K., and K. E. Kelly, Performance evaluation of the alphasense opc-n3 and plantower pms5003 sensor in measuring dust events in the salt lake valley, utah, *Atmospheric Measurement Techniques*, *16*(10), 2455–2470, doi:10.5194/amt-16-2455-2023, 2023.